# Execution Guided Line-by-Line Code Generation

**Boaz Lavon**        **Shahar Katz**        **Lior Wolf**
Blavatnik School of Computer Science and AI, Tel Aviv University
{boazlavon@mail, shaharkatz3@mail, wolf@cs}.tau.ac.il

## Abstract

We present a novel approach to neural code generation that incorporates real-time execution signals into the language model generation process. While large language models (LLMs) have demonstrated impressive code generation capabilities, they typically do not utilize execution feedback during inference, a critical signal that human programmers regularly leverage. Our method, Execution-Guided Classifier-Free Guidance (`EG-CFG`), dynamically incorporates execution signals as the model generates code, providing line-by-line feedback that guides the generation process toward executable solutions. `EG-CFG` employs a multi-stage process: first, we conduct beam search to sample candidate program completions for each line; second, we extract execution signals by executing these candidates against test cases; and finally, we incorporate these signals into the prompt during generation. By maintaining consistent signals across tokens within the same line and refreshing signals at line boundaries, our approach provides coherent guidance while preserving syntactic structure. Moreover, the method naturally supports native parallelism at the task level in which multiple agents operate in parallel, exploring diverse reasoning paths and collectively generating a broad set of candidate solutions. Our experiments across diverse coding tasks demonstrate that `EG-CFG` significantly improves code generation performance compared to standard approaches, achieving state-of-the-art results across various levels of complexity, from foundational problems to challenging competitive programming and data science tasks. Our code is available at: `https://github.com/boazlavon/eg_cfg`

## 1 Introduction

Large language models (LLMs) have recently demonstrated remarkable code generation capabilities, significantly advancing performance in tasks such as general programming problems [1, 2, 3], competitive coding challenges [4], and real-world software engineering tasks [5]. However, current LLM-based code generation methods primarily rely on pattern recognition derived from static representations of code rather than explicitly modeling code execution at runtime [1, 3]. Consequently, the generated programs often appear to be correct superficially, but fail to execute correctly on actual inputs, reflecting a critical gap between learned syntactic patterns and genuine executability.

State-of-the-art approaches typically use iterative refinement [6, 7, 8] or self-debugging strategies [9, 10]. Recent approaches adopt multi-agent or agentic workflows, explicitly employing iterative refinement and collaborative feedback mechanisms [11, 12, 13, 14]. However, these methods typically operate in discrete cycles: generating complete candidate solutions, executing them, and then using feedback from failures to guide subsequent attempts. Such approaches do not continuously integrate execution signals during inference, thus limiting their ability to dynamically adjust toward runtime correctness at the token level.

In contrast, human programmers frequently execute incomplete code fragments to quickly detect errors, assess progress, and iteratively refine their implementations based on concrete runtime outcomes, while exploring multiple candidate implementations and planning at varying levels of detail before finalizing solutions [15, 16]. This iterative, real-time refinement process explicitly

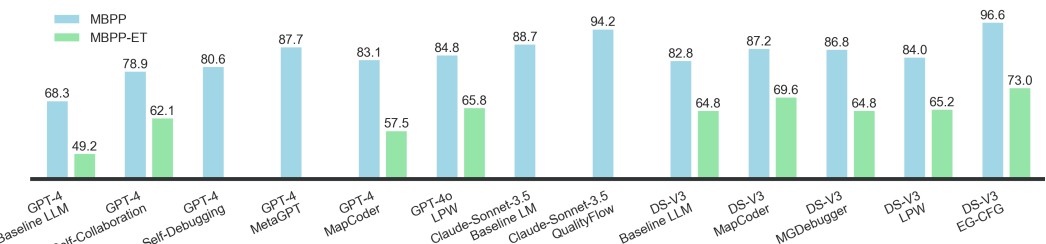

Figure 1: MBPP & MBPP-ET performance. `EG-CFG` (DeepSeek-V3) sets a new state-of-the-art results.

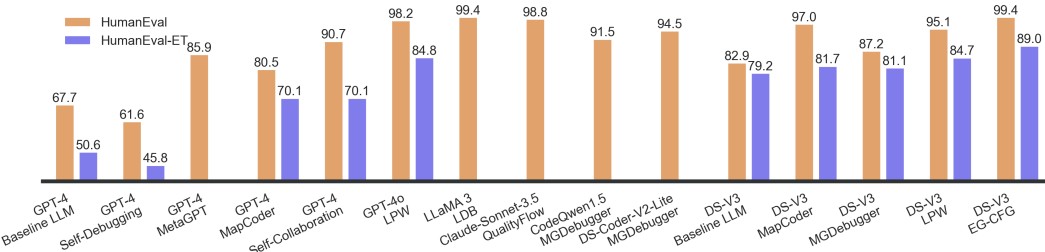

Figure 2: HumanEval & HumanEval-ET performance. `EG-CFG` (DeepSeek-V3) matches the state-of-the-art on HumanEval and sets a new state-of-the-art on HumanEval-ET.

grounds coding decisions in observed execution behavior rather than relying purely on syntactic or structural reasoning [17].

Inspired by these human coding practices and by LLM exploration techniques [18, 19, 20, 21], our proposed approach generates dynamic execution signals by explicitly sampling multiple candidate continuations at varied completion horizons, systematically adjusting decoding temperature to encourage exploration of different reasoning paths. Executing these diverse candidates yields rich execution-based feedback, explicitly mirroring human iterative refinement and exploratory problem-solving processes.

Unlike methods that provide explicit correctness indicators, such as scalar pass/fail or ranking signals [4, 22], or explicit verbal critiques and structured reflections on execution failures [9, 20], our method provides the raw execution outcome as a soft guidance signal. This approach allows the model to autonomously interpret and integrate minimally processed feedback into its generation process, bridging a significant gap between explicit externally-supervised reinforcement, in which the model is explicitly told what runs were successful [9], and implicit self-verification, where the model autonomously assesses the correctness of its own reasoning [10].

To incorporate the execution-based signal, our method utilizes Classifier-Free Guidance (CFG) [23], conditioning token-level generation decisions on the runtime outcome obtained by executing candidate code completions during inference. This approach guides the model toward solutions that are both syntactically plausible and executable, substantially improving correctness.

As depicted in Figure 1, our Execution-Guided Classifier-Free Guidance (`EG-CFG`) approach achieves state-of-the-art performance on the MBPP and MBPP-ET benchmarks [3], significantly outperforming existing methods. Using the open-source DeepSeek-V3-0324 model [24], `EG-CFG` attains 96.6% accuracy on MBPP and 73.0% on MBPP-ET, surpassing previous leading approaches such as QualityFlow [11] (94.2%), MetaGPT [25] (87.7%), and LPW [26] (84.8% on MBPP, 65.8% on MBPP-ET), all of which utilized leading closed-source models. On the HumanEval benchmark (Figure 2), `EG-CFG` achieves state-of-the-art accuracy of 99.4%, matching LDB [27] and establishes a new state-of-the-art on the HumanEval-ET benchmark, reaching 89.02% accuracy and surpassing LPW's 84.8% accuracy achieved with GPT-4o. Furthermore, `EG-CFG` establishes a new state-of-the-art on DS-1000 (Figure 3), a domain-specific benchmark focused on challenging data science problems, achieving 69.9% accuracy. Lastly, `EG-CFG` achieves a new state-of-the-art on the CodeContests benchmark (Figure 4) with 60.6% accuracy using DeepSeek-V3-0324, demonstrating its effectiveness on challenging competitive programming problems.

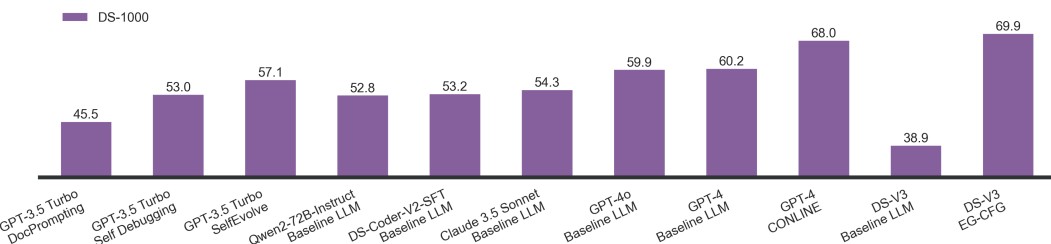

Figure 3: DS-1000 performance. `EG-CFG` (DeepSeek-V3) achieves new state-of-the-art, surpassing GPT-4.

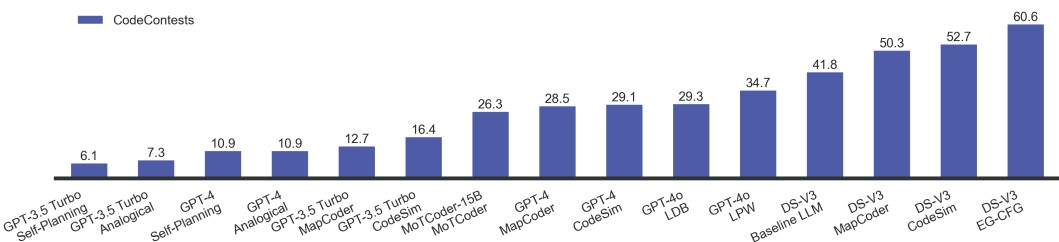

Figure 4: CodeContests performance. `EG-CFG` (DeepSeek-V3) sets a new state-of-the-art, outperforming GPT-4 and GPT-4o methods.

The superior performance of `EG-CFG` on MBPP-ET and HumanEval-ET highlights that our method not only generates accurate code but also significantly enhances robustness and reliability under complex and extended test scenarios. Summarizing, our main contributions are: (i) a framework for dynamically generating code while executing fragments of the code and using the execution traces to guide the generation process, (ii) introducing native parallelism at the task level that is not achievable in the sequential iterative refinement methods, (iii) using CFG in order to generate code that is conditioned on the execution feedback, and (iv) obtaining new state-of-the-art results on the MBPP, MBPP-ET, HumanEval-ET, DS-1000 and CodeContests benchmarks using open-source models, outperforming previous methods that are based on the leading closed-source models.

## 2 Related Work

Program synthesis has long served both as a means to evaluate LLMs' capabilities [1, 28] and as a key goal of the research community to automate code generation [29, 30]. Modern LLMs are typically assessed via automated benchmarks: given a coding problem description, the model generates code that is then validated on a set of test cases (unit-test). In this work, we focus on six code generation benchmarks: MBPP [3], HumanEval [1], DS-1000 [31] and CodeContests [4], along with the extended variants MBPP-ET and HumanEval-ET [32]. MBPP combines crowd-sourced tasks and math-based problems [33], HumanEval includes hand-written Python challenges, DS-1000 presents challenging data science problems involving libraries like Pandas and NumPy, and CodeContests features competitive programming problems requiring advanced algorithmic reasoning.

While zero-shot prompting, meaning directly querying an LLM on a task, is one evaluation strategy, few-shot prompting [34], which provides a small number of input–output examples, is more widely adopted. Other methods, such as Chain-of-Thought (CoT) [35], leverage the model's autoregressive nature to iteratively solve such tasks by decomposing them into sub-tasks. Similar approaches, such as Tree-of-Thoughts [21] extend CoT by exploring multiple candidate reasoning paths at a higher level of granularity.

With the growing ability of LLMs to utilize external signals and tools [36], LLMs have been augmented with feedback mechanisms to improve code generation. Approaches such as Reflexion [9], Program-aided Language Models (PAL) [6], and ReAct [7] iteratively update prompts based on external tools' outputs or intermediate results. In particular, Self-Debugging [10] demonstrates how LLMs can leverage debugging tools: the model generates code, tests it on available unit-test examples, and refines its output using the debugging feedback. Another related approach, LPW [26], employs a structured two-phase process involving initial high-level plan generation followed by

iterative refinement and debugging guided by execution feedback. The Multi-Granularity Debugger (MGDebugger) [37] also uses the LLM to simulate the execution of generated Python programs, employing the simulated trace as additional feedback for code refinement.

While some approaches use a single LLM to solve coding problems, agentic methods employ multiple LLM instances to tackle a single task. Works such as MetaGPT [25] introduce a framework in which a given task is split into multiple procedures, each assigned to a dedicated LLM agent. AgentCoder [12] extends these multi-agent frameworks by employing both a test-designer agent and an executor agent, effectively integrating the Self-Debugging paradigm [10] into an agentic workflow and demonstrating the benefits of execution feedback. Current agentic workflows, such as MapCoder [13] and LPW [26], typically rely on discrete cycles of sequential agent interactions or distinct refinement phases, inherently limiting their ability to fully leverage the power of parallelism within a single task. In contrast, `EG-CFG` breaks this barrier by introducing native parallelism at the task level. Multiple agents run concurrently on the same task, each with a different configuration, simultaneously exploring diverse reasoning paths without the constraints of sequential cycles, fully leveraging the power of parallelism.

Although code generation methods have advanced dramatically, they all rely on naive decoding methods such as temperature or top-p sampling [18]. In particular, execution-feedback techniques generate a complete solution, execute it, and then refine the code. For example, the model is prompted to produce a function, its implementation is tested against unit tests, and the resulting feedback is appended to the original prompt. This process repeats iteratively until a correct solution emerges. This approach contrasts with recent advances in LLM inference techniques, especially guidance methods such as Classifier Guidance [38] and Classifier-Free Guidance (CFG) [23, 39]. These methods condition generation on external constraints or classifier signals, directing the model to sample tokens from multiple contrastive distributions. Although CFG has shown strong performance, its guidance signals remain static, predefined and fixed throughout the sampling process [40].

As far as we can ascertain, guidance methods such as CFG have yet to be applied at scale to dynamic, execution-driven reasoning. This work bridges this gap by demonstrating that a single open-source model, augmented with execution feedback and a novel token-sampling strategy, outperforms the state-of-the-art across multiple widely adopted coding benchmarks at different complexity levels - from foundational problems to challenging competitive programming and data sceince tasks. Our contribution differs from prior work in multiple key aspects, including:

1. Prior approaches generate entire code blocks (or sub-blocks) and use the execution feedback of the entire block; by contrast, our approach gradually generates the solution line-by-line by sampling and executing multiple candidate continuation programs at each step.

2. Instead of explicitly relying on unit-test pass/fail signals or direct reflections on correctness, our method incrementally constructs code fragments and leverages their execution traces as an implicit feedback signal, guiding the model without explicit external supervision.

3. Our method relies on a single prompting scheme where multiple agents perform the same task and differ only in their parameter configurations. Each agent explores diverse reasoning paths and collectively generating a broad set of candidate solutions. This enables a level of parallelism that is unattainable in methods where agents communicate sequentially.

4. We replace naive decoding with CFG, using innovative trace-inspired prompts for dual-distribution interpolation sampling.

## 3 Method

We present Execution-Guided Classifier-Free Guidance (`EG-CFG`), a novel inference-time decoding method for neural code generation that explicitly integrates dynamic execution signals into the autoregressive generation process. A programming task $\tau$ is represented by three components:

$$\tau = (p_{\text{task}}, T, f_{\text{name}}), \tag{1}$$

where $p_{\text{task}}$ is a textual definition of the programming task, $T = \{t_j\}_{j=1}^{|T|}$ is the set of test cases, and $f_{\text{name}}$ is the target Python function name. The goal is to generate executable Python code $w^* = [w_0^*, w_1^*, \ldots, w_{N-1}^*]$ that solves the task correctly, formally satisfying:

$$\text{Execute}(w^*, t_j) = \text{success}, \quad \forall t_j \in T. \tag{2}$$

We consider two distinct instruction templates $p_0$: the standard DeepSeek-Coder 3-shot prompt [41], consisting of concise instructions accompanied by short illustrative examples, and an alternative

prompt explicitly designed to encourage the generation of step-by-step solutions with more atomic logic. See Appendix A for the example prompts used.

We build the instruction prompt by incorporating the task components into the prompt template:

$$p_{\text{inst}} = \text{BUILDINSTRUCTIONPROMPT}(p_0, \tau) \qquad (3)$$

At each step $i$, given a previously generated token sequence $w_{<i}$, the LLM $M$ assigns a probability distribution $M(w_i \mid p_{\text{inst},w_{<i}})$, which is conditioned on the instruction prompt $p_{\text{inst}}$. This generation proceeds iteratively until reaching a maximum token length $N_{\text{max}}$ or an end-of-code token. We build the instruction prompt by injecting the instruction task to the prompt template. We first run the model to produce an initial output sequence $p_{\text{pre}}$ based on $p_{\text{inst}}$:

$$w_{\text{pre}}^i = \arg\max M(w_i \mid p_{\text{inst},w_{<i}}), \qquad p_{\text{pre}} = [p_{\text{inst}}, w_{\text{pre}}^0, \cdots, w_{\text{pre}}^{n-1}] \qquad (4)$$

The generated sequence $p_{\text{pre}}$ contains reasoning tokens that precede the final executable solution. We identify the beginning of this solution block at index $i_{\text{solution}}$ by locating a special start-of-code token, which is defined in $p_0$. We then define the prefix of all preceding tokens, $p_{\text{pre-sol}}$, as:

$$p_{\text{pre-sol}} = p_{\text{pre}}[: i_{\text{solution}}] \qquad (5)$$

Additionally, we define an index $i_{\text{dyn}}$ that marks the end of the final few-shot example within the instruction prompt $p_{\text{inst}}$, which is used to inject future signals into the prompt.

The prompt $p_{\text{sol}}$ that we pass to the LLM is constructed autoregressively by aggregating executable code tokens into $p_{\text{pre-sol}}$ as formalized later in Equation 14 of section 3.3. This prompt contains partial solutions that are progressively updated until the solution is formed.

## 3.1 Dynamic Execution Feedback

Our dynamic execution feedback explicitly generates multiple candidate continuations based on a partially completed solution. Specifically, given $p_{\text{sol}}$ we generate a set of candidate continuations using beam-search decoding. Each candidate explicitly extends the current solution by $d$ additional lines of code, capturing meaningful variations in potential solutions and providing a granular basis for execution-based guidance. Formally, given parameters specifying the number of candidates $s$, completion horizon $d$, and sampling temperature $t$, the beam search sampling is performed to obtain $s$ candidates.

$$w_{c_j}^i \sim M(w_i \mid p_{\text{sol}}, w_{c_j}^{<i}; t) \text{ until } \text{CountLines}(c_j) \geq d, \ c_j = [w_{c_j}^0, \cdots, w_{c_j}^i], \ j = 1, \ldots, s \quad (6)$$

Each candidate $c^j$ is generated autoregressively with a stopping condition triggered after $d$ newline characters, representing a plausible continuation of the next $d$ lines of code. These candidates form the basis for generating detailed execution signals used to guide subsequent inference steps.

**Executable Extraction** To handle potentially invalid candidate continuations, we extract executable components via Abstract Syntax Tree (AST) parsing. Formally, for each candidate $c^j$, we apply the executable extraction function:

$$\hat{c}^j = \text{ExtractExecutable}(c^j), \quad j = 1, \ldots, s. \qquad (7)$$

This extraction function iteratively attempts to parse each candidate $c^j$ as follows: (1) attempt parsing $c^j$ (2) if unsuccessful, append a Python `pass` statement to the last line and retry; (3) if still unsuccessful, iteratively remove the last line and retry. This ensures minimal modifications for syntactic validity. After extraction, we apply uniqueness filtering to remove duplicates:

$$C = \text{Unique}\left(\{\hat{c}^j\}_{j=1}^s\right). \qquad (8)$$

This AST-based verification explicitly ensures that execution guidance relies solely on syntactically valid and executable code.

**Execution Feedback and Trace** For each unique executable candidate $\hat{c}^j \in C$, we explicitly execute it against all provided test cases $T = \{t_m\}_{m=1}^{|T|}$ and record the resulting execution feedback:

$$e^{j,m} = \text{ExtractExecutionFeedback}(\hat{c}^j, t_m), \quad \forall \hat{c}^j \in C, t_m \in T. \qquad (9)$$

Our approach is agnostic to the precise structure of the execution feedback $e^{j,m}$. In our implementation, execution feedback specifically takes the form of **execution traces:** a structured representation of a program's runtime behavior, capturing detailed step-by-step information during execution.

Specifically, we use a custom debugger for structured execution traces. A trace $e^{j,m}$ is defined as a sequence of $N^{j,m}$ runtime events, $\varepsilon_k$, resulting from the execution of a program $\hat{c}^j$ with input $t_m$:

$$e^{j,m} = [\varepsilon_k]_{k=1}^{N^{j,m}}, \quad \varepsilon_k = (E_k, \ell_k, v_k, \tau_k, r_k, x_k) \tag{10}$$

Each event $\varepsilon_k$ is a tuple containing the event type $E_k \in \{\texttt{call}, \texttt{line}, \texttt{return}, \texttt{exception}\}$, the source line number $\ell_k$, mappings from variable names to their values $(v_k)$ and types $(\tau_k)$, the return value $r_k$, and any exception details $x_k$. This structured representation provides comprehensive insight into the correctness and behavior of executed code, serving as a precise basis for dynamic feedback in our inference framework.

**Dynamic Signal Aggregation** The dynamic execution feedback signal is the concatenation of a fixed instruction string, denoted $p_{\text{dyn-inst}}$, to the aggregated execution feedback. This yields the dynamic signal prompt:

$$p_{\text{signal}} = [p_{\text{dyn-inst}}, \ \{(\hat{c}^j, t_m, e^{j,m})\}_{\hat{c}^j \in C, \, t_m \in T}], \tag{11}$$

where each tuple consists of a candidate completion $\hat{c}^j$, a test case $t_m$, and its corresponding execution trace $e^{j,m}$. Now we form a new prompt naming this prompt $p_{\text{dyn}}$ as **dynamic signal prompt**:

$$p_{\text{dyn}} = [p_{\text{sol}}[: i_{\text{dyn}}], p_{\text{signal}}, p_{\text{sol}}[i_{\text{dyn}} :]] \tag{12}$$

An example of the obtained prompt is shown in appendix Appendix B.

### 3.2 Classifier-Free Guidance (CFG)

Inspired by [23, 39], we utilize CFG to explicitly guide token generation by interpolating between two probability distributions: (i) an unconditional (prior) distribution based on the evolving solution prompt $p_{\text{sol}}$, and (ii) a conditional distribution based on dynamic signal prompt $p_{\text{dyn}}$ which incorporates execution feedback. Formally, for each token $w_i$, the CFG distribution is computed as:

$$\log M_{\text{CFG}}(w_i \mid p_{\text{sol}}, p_{\text{dyn}}) = \log M(w_i \mid p_{\text{sol}}) + \gamma \left[\log M(w_i \mid p_{\text{dyn}}) - \log M(w_i \mid p_{\text{sol}})\right], \tag{13}$$

where $\gamma \geq 0$ explicitly controls the strength of guidance. Higher values of $\gamma$ encourage the model to follow the execution-based guidance, while lower values allow greater flexibility toward the unconditional prior.

### 3.3 Execution-Guided Inference Loop

Our inference procedure extends standard autoregressive token generation by explicitly incorporating dynamic execution feedback via CFG. Starting from an initial prompt $p_{\text{pre-sol}}$ (Equation 5), we autoregressively sample tokens $w_i$ from the CFG-conditioned distribution $M_{\text{CFG}}(w_i \mid p_{\text{sol}}, p_{\text{dyn}})$, progressively constructing the solution sequence $p_{\text{sol}}$. At each token-generation step, we reuse the dynamic signal $p_{\text{signal}}$ (Equation 11), injecting it into $p_{\text{sol}}$ at index $i_{\text{dyn}}$ to form $p_{\text{dyn}}$ as described in Equation 12. $p_{\text{signal}}$ itself is regenerated only upon completing a new line.

$$w_{\text{sol}}^i = \arg\max M_{\text{CFG}}(w_i \mid p_{\text{sol}}, p_{\text{dyn}}), \qquad p_{\text{sol}} = [p_{\text{pre-sol}}, w_{\text{sol}}^0, \cdots, w_{\text{sol}}^{n-1}] \tag{14}$$

### 3.4 Parallel Multi-Agent Execution

Given an input task as defined in Equation 1, we launch a parallel multi-agent inference process where each agent is assigned a unique configuration: candidate count $s$, generation horizon $d$, sampling temperature $t$, instruction prompt template $p_0$ and guidance strength $\gamma$. Once any agent finds a correct solution, it is immediately returned and all remaining agents are terminated. The full pseudo-code for both the execution-guided inference loop and the multi-agent controller is provided in Appendix C.

## 4 Experiments

**Implementation Details** We conduct our experiments using two LLMs across different parameter scales: DeepSeek-Coder-1.3B [41], which is small enough to run locally on our machines (NVIDIA GeForce RTX 2080 Ti and RTX 3090 GPUs), and a large open-source model, DeepSeek-V3-0324 [24], which we use through a cloud inference endpoint. We use *Fireworks AI* which was chosen based on two criteria: a modest cost, and the availability of a log probability output that is required to perform the CFG (Equation 13). The full code used for our experiments is provided in the supplementary materials.

**Hyperparameter Settings** As explained in section 3.4, our method launches multiple parallel agents for each task. Each agent is assigned a different hyperparameter configuration. The following hyper-parameter sets were used in our experiments: $s = 3$, $t \in \{0.7, 0.75, 0.85, 0.95, 1.2, 1.5\}$, $d \in \{2, 3, 6, 8\}$, $\gamma \in \{0, 0.5, 1, 3\}$. Additionally, we evaluate both $p_0$ prompt templates, see section 3 and appendix Appendix A.

**Evaluation Benchmark** Our evaluations use widely-adopted benchmarks: MBPP [3] (500 tasks) and HumanEval [1] (164 tasks), along with their extended test versions MBPP-ET and HumanEval-ET [32]. To assess performance on more challenging tasks, we also evaluate on the DS-1000 data science benchmark [31] (1000 tasks) and the CodeContests competitive programming benchmark [4] (using the ExecEval framework [42]). We report accuracy: the percentage of problems passing all test cases. To rigorously test generalization and prevent overfitting to public tests, evaluations on HumanEval, HumanEval-ET, MBPP-ET, CodeContests, and DS-1000 rely on hidden test cases inaccessible during inference. DS-1000's structure, providing only a single input-output example per problem, further tests model robustness against reliance on examples.

**Baselines** We compare our `EG-CFG` method against several established state-of-the-art methods for code generation and debugging: a Baseline LLM using the few-shot template from DeepSeek-Coder's evaluation [41]; MGDebugger [37], an iterative refinement method combining test-case feedback with LLM-simulated execution traces; MapCoder [13], which employs multi-agent interactions; QualityFlow [11], which incorporates agentic workflows for iterative enhancement; LPW [26], which utilizes a structured two-phase workflow with runtime execution feedback; and CodeSim [14], a search-based method that adapts semantically similar code snippets.

## 4.1 Results

Using the DeepSeek-V3-0324 model, EG-CFG achieves new state-of-the-art (SOTA) results across all evaluated benchmarks. On MBPP and MBPP-ET (Table 1), it surpasses prior methods using large closed-source models like GPT-4 and Claude 3.5. On HumanEval (Table 2), it matches the state-of-the-art (LDB) at 99.4% and sets a new state-of-the-art on the challenging HumanEval-ET variant. Furthermore, it establishes new state-of-the-art on CodeContests (Table 3), significantly exceeding its own baseline, MapCoder, and previous GPT-4 based approaches like LPW and CodeSim, and also sets a new state-of-the-art on DS-1000 (69.9%, Table 4), outperforming CONLINE/GPT-4.

We note that across all tested baselines, the publicly available code was highly sensitive to the specific model and could not be readily applied to DeepSeek models. We invested substantial effort in debugging and adapting the code to ensure it produced meaningful results that represented each baseline method as favorably as possible. Other methods, such as QualityFlow, have not released their code, preventing us from evaluating them on the DeepSeek models. While LPW did release a public implementation, we encountered substantial technical issues during the execution of their published code on the DeepSeek-Coder 1.3b model, resulting in unusually low scores despite our best debugging efforts on that benchmark.

**Run time** A key advantage of EG-CFG is its native parallelism, a design choice that aligns with the growing trend of leveraging large-scale, parallel compute in modern agentic systems. This architectural choice is a key differentiator from iterative refinement methods, which are inherently sequential and therefore cannot leverage parallel compute to accelerate work on a single task. In contrast, EG-CFG is designed to explore diverse reasoning paths simultaneously across multiple agents (as described in section 3.4). This fundamental architectural difference makes wall-clock time the most relevant and fair metric for comparison. As can be seen in Table 5, on the MBPP benchmark, our method is more efficient than MGDebugger and competitive with MapCoder. When comparing with LPW, our method is more efficient with the smaller model but slower with the larger model.

**Fairness of Comparison** To demonstrate EG-CFG's qualitative advantage is not just a larger compute budget, we drastically increased baseline token usage. On MBPP (DeepSeek-V3-0324), we raised MGDebugger and MapCoder retry counts 40-fold (from 5 to 200). Despite this 40x compute increase, gains were marginal : MGDebugger improved by only 6.8 percentage points, from 86.8% to 93.6% (solving 34 of its 86 prior failures). MapCoder's improvement was even smaller, at 1.6 percentage points, from 87.2% to 88.8% (solving 8 of its 64 prior failures). Both

Table 1: Performance on the MBPP and MBPP-ET benchmarks. Our proposed `EG-CFG` achieves a new state-of-the-art overall accuracy. The DeepSeek–Coder-1.3B and –V3-0324 results for all baselines were obtained by our study using the official implementations provided by each baseline method. The results below the double separator were collected from the respective papers.

| Model | Method | MBPP | | MBPP-ET | |
|---|---|---|---|---|---|
| | | Acc. (%) | RSR (%) | Acc. (%) | RSR (%) |
| DeepSeek-Coder 1.3B | Baseline LLM | 49.4 | 0.0 | 42.6 | 0.0 |
| DeepSeek-Coder 1.3B | EG-CFG (Ours) | 83.2 | 66.79 | 59.8 | 29.96 |
| DeepSeek-Coder 1.3B | MapCoder [13] | 55.2 | 11.46 | 46.2 | 6.27 |
| DeepSeek-Coder 1.3B | MGDebugger [37] | 70.4 | 41.5 | 44.6 | 3.48 |
| DeepSeek-V3-0324 | Baseline LLM | 82.8 | 0.0 | 64.8 | 0.00 |
| DeepSeek-V3-0324 | EG-CFG (Ours) | **96.6** | **80.23** | **73.0** | **23.29** |
| DeepSeek-V3-0324 | MapCoder [13] | 87.2 | 25.58 | 69.6 | 13.63 |
| DeepSeek-V3-0324 | MGDebugger [37] | 86.8 | 23.25 | 64.8 | 0.00 |
| DeepSeek-V3-0324 | LPW [26] | 84.0 | 6.97 | 65.2 | 1.13 |
| GPT-4 | Baseline LLM | 68.3 | - | 49.2 | - |
| GPT-4 | Self-Collaboration [43] | 78.9 | - | 62.1 | - |
| GPT-4 | Self-Debugging [10] | 80.6 | - | - | - |
| GPT-4 | MetaGPT [25] | 87.7 | - | - | - |
| GPT-4 | MapCoder [13] | 83.1 | - | 57.5 | - |
| GPT-4o | LPW [26] | 84.8 | - | 65.8 | - |
| CodeQwen1.5 | MGDebugger [37] | 80.8 | - | - | - |
| DeepSeek-Coder-V2-Lite | MGDebugger [37] | 80.0 | - | - | - |
| Claude-Sonnet-3.5 | Baseline LLM [11] | 88.7 | - | - | - |
| Claude-Sonnet-3.5 | QualityFlow [11] | 94.2 | - | - | - |

Table 2: Performance on the HumanEval and HumanEval-ET benchmarks. Our proposed `EG-CFG` matches the state-of-the-art on HumanEval and achieves a new state-of-the-art on HumanEval-ET.

| Model | Method | HumanEval | | HumanEval-ET | |
|---|---|---|---|---|---|
| | | Acc. (%) | RSR (%) | Acc. (%) | RSR (%) |
| DeepSeek-V3-0324 | Baseline LLM | 82.92 | 0.0 | 79.20 | 0.0 |
| DeepSeek-V3-0324 | EG-CFG (Ours) | **99.4** | **94.04** | **89.02** | **47.21** |
| DeepSeek-V3-0324 | MapCoder [13] | 96.95 | 82.14 | 81.70 | 12.02 |
| DeepSeek-V3-0324 | MGDebugger [37] | 87.20 | 25.05 | 81.09 | 9.09 |
| DeepSeek-V3-0324 | LPW [26] | 95.12 | 71.42 | 84.74 | 26.63 |
| GPT-4 | Baseline LLM | 67.7 | - | 50.6 | - |
| GPT-4 | Self-Collaboration [43] | 90.7 | - | 70.1 | - |
| GPT-4 | Self-Debugging [10] | 61.6 | - | 45.8 | - |
| GPT-4 | MetaGPT [25] | 85.9 | - | - | - |
| GPT-4 | MapCoder [13] | 80.5 | - | 70.1 | - |
| GPT-4o | LPW [26] | 98.2 | - | 84.8 | - |
| LLaMA 3 | LDB [27] | 99.4 | - | - | - |
| Claude-Sonnet-3.5 | QualityFlow [11] | 98.8 | - | - | - |
| CodeQwen1.5 | MGDebugger [37] | 91.5 | 64.1 | - | - |
| DeepSeek-Coder-V2-Lite | MGDebugger [37] | 94.5 | 76.3 | - | - |

remained significantly below `EG-CFG`'s 96.6% accuracy , showing our performance gap is due to the execution-guided feedback loop's quality, rather than the amount of computation.

**Ablation study** We performed an ablation study on the MBPP and MBPP-ET benchmarks to evaluate various components of our method. The results are reported in Table 6. When omitting the beam search of Equation 6, which creates multiple solutions instead of a single completion, the performance of the method drops and becomes much closer to the baseline performance. The role of CFG is evident from the second ablation, in which a value of $\gamma = 1$ is used in Equation 13. In this case, there is a clear drop in performance, although results are still clearly above the baseline. Finally, a similar drop in performance is observed when replacing the detailed execution trace used as part of the dynamic signal with a minimal execution trace, formally defined as the final event of the execution trace (Equation 10).

Table 3: Performance on the CodeContests benchmark. Our proposed `EG-CFG` achieves a new state-of-the-art overall accuracy on CodeContests. The runs on DeepSeek-V3-0324 are by us, and all other results are quoted from the literature. * The LPW results were obtained on a custom test-set; the published code was not compatible with evaluating on DeepSeek. **Reported by the MapCoder paper [13]. ***Reported by the CodeSim paper [14].

| Model | Method | Accuracy (%) | RSR (%) |
|---|---|---|---|
| DeepSeek-V3-0324 | Baseline LLM | 41.81 | 0.00 |
| DeepSeek-V3-0324 | EG-CFG (Ours) | **60.6** | **32.29** |
| DeepSeek-V3-0324 | MapCoder [13] | 50.30 | 14.59 |
| DeepSeek-V3-0324 | CodeSim [14] | 52.72 | 18.76 |
| GPT-4o | LPW [26]* | 34.7 | - |
| GPT-4o | LDB [27]*** | 29.3 | - |
| GPT-4 | CodeSim [14] | 29.1 | - |
| GPT-4 | MapCoder [13] | 28.5 | - |
| GPT-4 | Self-Planning [44]** | 10.9 | - |
| GPT-4 | Analogical [45]** | 10.9 | - |
| GPT-3.5 Turbo | CodeSim [14] | 16.4 | - |
| GPT-3.5 Turbo | MapCoder [13] | 12.7 | - |
| GPT-3.5 Turbo | Analogical [45]** | 7.3 | - |
| GPT-3.5 Turbo | Self-Planning [44]** | 6.1 | - |
| MoTCoder-15B | MoTCoder [46] | 26.34 | - |

Table 4: Performance on the DS-1000 benchmark. Our proposed `EG-CFG` achieves a new state-of-the-art overall accuracy on DS-1000. *Reported by the SelfEvolve paper [47]. **Reported by DS-1000 [31] official leaderboard.

| Model | Method | Accuracy (%) | RSR (%) |
|---|---|---|---|
| DeepSeek-V3-0324 | EG-CFG (Ours) | **69.9** | **50.73** |
| DeepSeek-V3-0324 | Baseline LLM | 38.9 | 0.00 |
| GPT-4 | CONLINE [48] | 68.0 | - |
| GPT-4 | Baseline LLM | 60.2 | - |
| GPT-3.5 Turbo | SelfEvolve [47] | 57.1 | - |
| GPT-3.5 Turbo | Self Debugging [10]* | 53.0 | - |
| GPT-3.5 Turbo | DocPrompting [49]* | 45.50 | - |
| GPT-4o | Baseline LLM** | 59.9 | - |
| Claude 3.5 Sonnet | Baseline LLM** | 54.3 | - |
| DeepSeek-Coder-V2-SFT | Baseline LLM** | 53.2 | - |
| Qwen2-72B-Instruct | Baseline LLM** | 52.8 | - |

## 5 Conclusions

This paper introduces Execution-Guided Classifier-Free Guidance (`EG-CFG`), a novel approach that fundamentally reframes neural code generation by incorporating real-time execution signals directly into the inference process. Our method bridges a critical gap between static pattern recognition and execution semantics by dynamically sampling candidate continuations, extracting execution traces, and leveraging these signals to guide token-level generation decisions.

The empirical results demonstrate that `EG-CFG` achieves new state-of-the-art results across a diverse set of benchmarks. Using DeepSeek-V3-0324, it achieves 96.6% accuracy on MBPP, 73.0% on MBPP-ET, 99.4% on HumanEval, 89.02% on HumanEval-ET, 69.9% on DS-1000, and 60.6% on CodeContests, surpassing both open-source and proprietary model-based approaches. The superior performance on both MBPP-ET and HumanEval-ET underscores the method's robustness and effectiveness in generating reliable and accurate code under complex and extended test scenarios. Notably, our approach demonstrates robust performance even with smaller models, achieving 83.2% accuracy using DeepSeek-Coder-1.3B, comparable to results from substantially larger models like GPT-4. This scalability highlights the effectiveness of execution signals as a guiding mechanism regardless of model capacity.

The `EG-CFG` approach offers several advantages over existing methods. Unlike discrete iterative refinement techniques that operate at coarse granularity between complete solution attempts, our method provides continuous feedback at the token level. By integrating execution signals that reflect

Table 5: Per-task runtime statistics (in seconds) for each model and method on MBPP.

| Model | Method | Mean ± SD (s) |
|---|---|---|
| DeepSeek-Coder 1.3b | EG-CFG | 123.23 ± 344.91 |
| DeepSeek-Coder 1.3b | MGDebugger | 495.16 ± 411.07 |
| DeepSeek-Coder 1.3b | MapCoder | 121.9 ± 213.89 |
| DeepSeek-Coder 1.3b | LPW | 197.71 ± 128.07 |
| DeepSeek-V3-0324 | EG-CFG | 271.37 ± 271.45 |
| DeepSeek-V3-0324 | MGDebugger | 842.24 ± 705.19 |
| DeepSeek-V3-0324 | MapCoder | 283.84 ± 197.54 |
| DeepSeek-V3-0324 | LPW | 87.51 ± 210.84 |

Table 6: Ablation results for EG-CFG on DeepSeek-Coder 1.3b on MBPP and MBPP-ET benchmarks.

| Method | MBPP | | MBPP-ET | |
|---|---|---|---|---|
| | Acc. (%) | RSR (%) | Acc. (%) | RSR (%) |
| EG-CFG | 83.2 | 66.79 | 59.8 | 29.96 |
| EG-CFG, no beam search | 58.2 | 17.39 | 43.6 | 1.74 |
| EG-CFG w/o CFG ($\gamma = 1$) | 75.2 | 50.98 | 48.2 | 9.74 |
| EG-CFG, minimal trace | 76.4 | 53.35 | 51.2 | 14.98 |
| Baseline LLM | 49.4 | 0.0 | 42.6 | 0.0 |

actual runtime behavior, EG-CFG mirrors the incremental testing and debugging process that human programmers employ.

Looking forward, this work opens several promising research directions. The execution-guided framework could be extended to more complex programming tasks requiring longer-horizon planning or multi-file interactions. Additionally, the principles of EG-CFG, which dynamically incorporate external semantic signals into generation, could benefit domains that rely on grounding in systems, such as database querying, formal verification, or simulation-based tasks. More broadly, EG-CFG represents a shift in generative modeling beyond static, pattern-based generation toward responsive, context-aware methods informed by environmental interaction. As models scale and applications expand, such approaches can drive the development of systems that generate high-quality outputs while reasoning about their real-world behavior, enabling more reliable and aligned generation.

## 6 Limitations

While the EG-CFG framework demonstrates significant improvements in code generation performance, several important limitations should be acknowledged.

First, the approach introduces computational overhead compared to standard inference methods. The beam search exploration, execution of multiple candidate continuations, and dual-distribution interpolation in the CFG mechanism collectively increase inference time. Though our parallel execution strategy mitigates this overhead somewhat, future work should explore more efficient methods for extracting and incorporating execution signals. Second, EG-CFG's effectiveness is contingent upon the availability of executable test cases that adequately exercise the target functionality. In real-world programming scenarios, comprehensive test cases may not always be available or easily generated by LLMs, potentially limiting the approach's applicability.

Finally, because our inference loop is bottom-up, it does not exploit task decomposition, a strategy that has been shown to improve code generation [37]. Future work could integrate our sampling strategy with iterative refinement methods, task-decomposition methods or with top-down problem-inspection techniques [50] to achieve even better performance. Despite these limitations, EG-CFG represents a significant advancement in execution-aware code generation and provides a solid foundation for future research addressing these challenges.

## Acknowledgements

This work was supported by a Tel Aviv University Center for AI and Data Science (TAD) grant. This research was also supported by the Ministry of Innovation, Science & Technology, Israel (1001576154) and the Michael J. Fox Foundation (MJFF-022407). The contribution of SK is part of a PhD thesis research conducted at Tel Aviv University.

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

# A  The two $p_{\text{inst}}$ prompts used by our method

The two basic instruction prompts used by our method are provided in Figure 5 and Figure 6. The former is the standard DeepSeek-Coder 3-shot prompt [41], consisting of concise instructions accompanied by short illustrative examples, and the latter is an alternative prompt explicitly designed to encourage the generation of step-by-step solutions with more atomic logic.

## Raw DeepSeek-Instruct Prompt for MBPP Task 395

```
You are an AI programming assistant, utilizing the Deepseek Coder model,
developed by Deepseek Company, and you only answer questions related to
computer science.
### Instruction:
Please refer the given examples and generate a python function for my problem.
Examples are listed as follows:
- Example 1:
>>> Problem:
Write a function to find the similar elements from the given two tuple lists.
>>> Test Cases:
assert similar_elements((3, 4, 5, 6),(5, 7, 4, 10)) == (4, 5)
assert similar_elements((1, 2, 3, 4),(5, 4, 3, 7)) == (3, 4)
assert similar_elements((11, 12, 14, 13),(17, 15, 14, 13)) == (13, 14)

>>> Code:
def similar_elements(test_tup1, test_tup2):
  res = tuple(set(test_tup1) & set(test_tup2))
  return (res)

- Example 2:
>>> Problem:
Write a python function to identify non-prime numbers.
>>> Test Cases:
assert is_not_prime(2) == False
assert is_not_prime(10) == True
assert is_not_prime(35) == True

>>> Code:
import math
def is_not_prime(n):
    result = False
    for i in range(2,int(math.sqrt(n)) + 1):
        if n % i == 0:
            result = True
    return result

- Example 3:
>>> Problem:
Write a function to find the largest integers from a given list of numbers using...
>>> Test Cases:
assert heap_queue_largest([25, 35, 22, 85, 14, 65, 75, 22, 58],3) == [85, ...
assert heap_queue_largest([25, 35, 22, 85, 14, 65, 75, 22, 58],2) == [85, ...
assert heap_queue_largest([25, 35, 22, 85, 14, 65, 75, 22, 58],5) == [85, ...

>>> Code:
import heapq as hq
def heap_queue_largest(nums,n):
  largest_nums = hq.nlargest(n, nums)
  return largest_nums

Here is my problem:
>>> Problem:
Write a python function to find the first non-repeated character in a given
string.
>>> Test Cases:
assert first_non_repeating_character("abcabc") == None
assert first_non_repeating_character("abc") == "a"
assert first_non_repeating_character("ababc") == "c"

### Response:
```

Figure 5: The DeepSeek-Instruct prompt used for MBPP Task 395. This prompt includes multiple solved examples followed by the target task.

```
You are an AI programming assistant, utilizing the Deepseek Coder model,
developed by Deepseek Company, ...
### Instruction:
Write a python function to find the first non-repeated character in a given
string.

Write a Python function that satisfies the following test cases:
>>> Test Cases:
['assert first_non_repeating_character("abcabc") == None',
 'assert first_non_repeating_character("abc") == "a"',
 'assert first_non_repeating_character("ababc") == "c"']

Your solution should be written in as many lines as possible.
This ensures that prefixes of your function remain valid Python programs.
Allowing **incremental execution and debugging**.

Write the function **step by step**, progressively introducing variables and logic.
Avoid using list comprehensions, lambda functions, or overly compact one-liners.
Instead, follow these guidelines:**

Avoid list comprehensions, use loops instead:
Incorrect:
def square_numbers(lst):
    return [x ** 2 for x in lst]

Correct:
def square_numbers(lst):
    squares = []
    for num in lst:
        squared_value = num ** 2
        squares.append(squared_value)
    return squares

Avoid inline expressions, use variables instead
Incorrect:
def calculate_area(length, width):
    return (length * width) / 2
Correct:
def calculate_area(length, width):
    product = length * width
    area = product / 2
    return area
Incorrect:
result.append(x + y)
Correct:
z = x + y
result.append(z)
Incorrect:
def compute_value(a, b, c):
    return (a + b) * (c / (a - b) + (a * c) / (b + c))
Correct:
def compute_value(a, b, c):
    term1 = a + b
    term2 = a - b
    term3 = c / term2
    term4 = a * c / (b + c)
    result = term1 * (term3 + term4)
    return result

### Response:
```

Figure 6: Long-Instruct prompt for MBPP Task 395. This instruction-only prompt includes stylistic constraints that encourage traceable, step-by-step completions suitable for dynamic signal extraction.

# B Example Dynamic Prompt Structure

This appendix provides a concrete example illustrating the structure of the dynamic prompt $p_{dyn}$ used by our EG-CFG method (described at the end of Section 3.1). Figure 7 below shows how the different components are assembled into the dynamic prompt $p_{dyn}$. It displays the original task instruction ($p_{inst}$), followed by the dynamically injected signal ($p_{signal}$). This signal begins with the dynamic signal instruction ($p_{dyn-inst}$) and includes execution traces for each candidate, obtained in this example by executing $s = 2$ candidates against a single test case ($|T| = 1$). Finally, the figure shows the model's partially completed response generated up to that point.

**Instruction from** $p_{\text{inst}}$: Write a python function to find the first non-repeated character in a given string.
```
assert first_non_repeating_character("aabc") == "b"
assert first_non_repeating_character("abcabc") == None
assert first_non_repeating_character("abc") == "a"
 assert first_non_repeating_character("ababc") == "c"
```

**Dynamic Signal Instruction ($p_{\text{dyn-inst}}$) starts at $i_{\text{dyn}}$**
Below are execution traces from running the response function after appending several possible future continuations. These continuations represent plausible ways the function might continue from its current state. They are not necessarily full solutions - some may be partial, exploratory, or incomplete. For each candidate continuation, multiple test cases (invocations) were executed to observe its behavior under different inputs. Each entry includes:
  - A candidate version of the function
  - A specific test case used for invocation
  - The resulting execution trace for that test case
These dynamic signals can help you better understand how different plausible continuations behave at runtime, and guide you toward a more accurate solution.

**Execution feedback for a single test ($|T| = 1$) and $s = 2$ candidates:**

```
# Function:
```
```python
def first_non_repeating_character (s):
    char_count = {}
    for char in s:
        char_count [char] = char_count.get(char, 0) + 1
    for char in s:
        if char_count [char] == 1:
            return char
    return None
```

```
# Invocation: first_non_repeating_character("aabc")
# Execution Trace:
```
```
s = 'aabc', char_count = {}
...
char = 'c' -> char_count = {'a': 2, 'b': 1, 'c': 1}
char = 'b' -> count = 1 -> return 'b'
```

```
# Function:
```
```python
def first_non_repeating_character (s):
    char_count = {}
    for char in s:
        char_count [char] = char_count.get(char, 0) + 1
    for char in s:
        if char_count [char] == 2:
            return char
    return None
```

```
# Invocation: first_non_repeating_character("aabc")
# Execution Trace:
```
```
s = 'aabc', char_count = {}
...
char = 'c' -> char_count = {'a': 2, 'b': 1, 'c': 1}
char = 'a' -> count = 2 -> return 'a'
```

```
### Response:
```
```python
def first_non_repeating_character (s):
    char_count = {}
    for char in s:
        char_count [char] = char_count.get(char, 0) + 1
```

Figure 7: Example of $p_{\text{dyn}}$ with injected $p_{\text{signal}}$ at index $i_{\text{dyn}}$

.

## C   `EG-CFG` **Pseudo-Code**

To complement the formal description in Section 3, we provide full pseudo-code for our method in this appendix. Algorithm 1 outlines the controller responsible for coordinating a multi-agent parallel inference process, where each agent explores diverse reasoning paths using a unique configuration. Each agent independently invokes the core decoding routine defined in Algorithm 2, which integrates dynamic execution feedback via classifier-free guidance (CFG). This loop incrementally generates a solution by incorporating real-time execution feedback into the prompt.

---

**Algorithm 1** Multi-Agent Parallel Inference Controller

---

**Input:** Task $\tau = (p_{\text{task}}, T, f_{\text{name}})$ (Eq.1), Model $M$, `EG-CFG` Configurations $\mathcal{H} = \{(p_0, s, d, t, \gamma)\}$
1: Launch parallel agents $A = \{a_h : h \in \mathcal{H}\}$
2: **for all** agents $a_h \in A$ with config $(p_0, s, d, t, \gamma)$ in parallel **do**
3:     $p_{\text{inst}} \leftarrow$ BUILDINSTRUCTIONPROMPT$(p_0, \tau)$                   ▷ Build instruction prompt, Equation 3
4:     $p_{\text{sol}} \leftarrow$ EG-CFG-INFERENCELOOP$(p_{\text{inst}}, T, M, s, d, t, \gamma)$
5:     **if** Execute$(p_{\text{sol}}, t_j) = \text{success}, \forall t_j \in T$ **then**                   ▷ Verify solution, Equation 2
6:         **return** $p_{\text{sol}}$                   ▷ Return first valid solution and terminate other agents
7:     **end if**
8: **end for**
9: **return** `null`                   ▷ Failure: No correct solution found

---

**Algorithm 2** Execution-Guided Classifier-Free Guidance (EG-CFG) Inference Loop

---

**Input:** $p_{\text{inst}}, T, M, s, d, t, \gamma$
1: $p_{\text{pre}} \leftarrow M(p_{\text{inst}})$                   ▷ Generate initial output sequence, Equation 4
2: Locate $i_{\text{solution}}, i_{\text{dyn}}$ in $p_{\text{pre}}$
3: $p_{\text{pre-sol}} \leftarrow p_{\text{pre}}[: i_{\text{solution}}]$                   ▷ Extract reasoning prefix before solution code, Equation 5
4: $p_{\text{sol}} \leftarrow p_{\text{pre-sol}}$
5: $p_{\text{signal}} \leftarrow$ `null`
6: **while** true **do**
7:     **if** $p_{\text{signal}} = $ `null` or ($w_{i-1}$ exists and `'\n'` in $w_{i-1}$) **then**
8:         $\mathcal{C}_{raw} \leftarrow []$                   ▷ Initialize list for executable candidates
9:         **for all** each $j = 1, \ldots, s$ **do**
10:             Initialize $c_j \leftarrow []$
11:             **while** CountLines$(c_j) < d$ **do**                   ▷ Generate a candidate continuation, Equation 6
12:                 Sample $w_{c_j}^k \sim M(w_k \mid p_{\text{sol}}, w_{c_j}^{<k}; t)$
13:                 Append $w_{c_j}^k$ to $c_j$
14:             **end while**
15:             $\hat{c}^j \leftarrow$ ExtractExecutable$(c^j)$                   ▷ Extract executable part, Equation 7
16:             Append $\hat{c}^j$ to $\mathcal{C}_{raw}$
17:         **end for**
18:         $C \leftarrow$ Unique$(\mathcal{C}_{raw})$                   ▷ Filter for unique candidate continuations, Equation 8
19:         $\mathcal{E} \leftarrow []$                   ▷ Initialize list for feedback tuples
20:                           ▷ Candidate executions against test cases run in parallel, Equation 9
21:         **for all** each $\hat{c}^j \in C$ **do**
22:             **for all** each $t_m \in T$ **do**
23:                 $e^{j,m} \leftarrow$ ExtractExecutionFeedback$(\hat{c}^j, t_m)$
24:                 Append $(\hat{c}^j, t_m, e^{j,m})$ to $\mathcal{E}$
25:             **end for**
26:         **end for**
27:         $p_{\text{signal}} \leftarrow [p_{\text{dyn-inst}}, \mathcal{E}]$                   ▷ Construct the full dynamic signal, Equation 11
28:     **end if**
29:     $p_{\text{dyn}} \leftarrow [p_{\text{sol}}[: i_{\text{dyn}}], p_{\text{signal}}, p_{\text{sol}}[i_{\text{dyn}} :]]$                   ▷ Inject signal into the prompt, Equation 12
30:     $w_i \leftarrow \arg\max M_{\text{CFG}}(w \mid p_{\text{sol}}, p_{\text{dyn}})$                   ▷ Sample next token using CFG, Eq.13 Eq.14
31:     Append $w_i$ to $p_{\text{sol}}$                   ▷ Append new token to the solution, Eq.13 Eq.14
32:     **if** $w_i = $ `EOT` or length$(p_{\text{sol}}) \geq N_{\text{max}}$ **then**
33:         **return** $p_{\text{sol}}$
34:     **end if**
35: **end while**

---

