# OpenReview forum: "Execution Guided Line-by-Line Code Generation"
_NeurIPS.cc/2025/Conference — NeurIPS 2025 poster_

### Official Review · Reviewer_GpF1 · 2025-06-21

**Clarity:** 3
**Significance:** 3
**Originality:** 3
**Rating:** 5
**Confidence:** 3

**Summary:**

The paper proposes an interesting program synthesis approach called **EG-CFG**, which provides the language model with real-time, line-by-line execution feedback of current variables. During inference-time decoding, EG-CFG generates the next token by aggregating the distributions from two sources: the original prompt without inline execution feedback and a dynamic prompt enriched with inline execution feedback. The method is evaluated on a programming by examples with language instructions benchmark: the MBPP benchmark. The EG-CFG method achieves state-of-the-art accuracy performance (96.6%) on MBPP, significantly outperforming other baselines using the latest proprietary LLMs.

**Questions:**

- How does the computational cost (e.g. API-call tokens) of the EG-CFG method compare with other baselines? If the costs differ significantly, the comparison may not be fair.
- The MBPP dataset contains relatively simple tasks with few variables. How does the method perform when applied to problems with more variables and inputs? This could increase the complexity of inline execution feedback and potentially result in an overly long prompt.

**Ethical Concerns:**

["NO or VERY MINOR ethics concerns only"]

**Final Justification:**

I am raising my score from 4 to 5 for the following reasons:

1.	The program synthesis approach with inline program execution feedback is novel, effective, and elegant.

2.	The original paper only conducted experiments on a single programming domain (MBPP), but the authors have now expanded the evaluation and achieved state-of-the-art performance across all major benchmarks: MBPP (96.6%), MBPP-ET (73.0%), HumanEval (99.4%), HumanEval-ET (89.0%), CodeContests (60.6%), and DS-1000 (69.9%). This demonstrates that the method generalizes well across diverse coding tasks.

3.	The method remains effective on complex tasks involving more variables, longer code, and limited context windows—addressing my initial concern that it may only work on simple tasks with short code.

**Limitations:**

yes

**Paper Formatting Concerns:**

good

**Quality:**

3

**Strengths And Weaknesses:**

**Strengths**

- **Quality:** The paper introduces a novel program synthesis approach with inline program execution feedback, mimicking how humans write and debug Python code using breakpoints. This strategy effectively increases the number of tasks successfully solved.
- **Significance:** The proposed method achieves new state-of-the-art performance on the MBPP benchmark (96.6%), and outperforms existing approaches across both 671B and 1.3B model sizes.
- **Clarity:** The paper is well-written and easy to follow. The method is clearly explained through both natural language and mathematical formulations, and the prompt example provides a helpful illustration.

**Weakness**

- **Significance:** The method is only evaluated on a single programming domain, MBPP. It would strengthen the paper to include results on more diverse and challenging datasets to demonstrate broader applicability.
- **Quality:** Program synthesis methods often benefit from increased sampling due to the ability to verify outputs against test cases. It would be helpful to include a plot showing how the method's accuracy performance scales with greater inference-time compute (e.g., more samples).

---

> ### Author Rebuttal · Authors · 2025-07-31
>
> Thank you for your encouraging and thoughtful review. We particularly appreciate your recognition of the method’s novelty and the clarity of its presentation. Your feedback has been highly motivating, and we’ve made several meaningful revisions to further improve the paper. Below, we address your comments point by point.
> ## Significance: Benchmark Diversity
> We have substantially expanded our evaluation beyond MBPP to address concerns about benchmark diversity and broader applicability.
> Our updated experiments include the additional five benchmarks:
> - **HumanEval**: A foundational and widely adopted benchmark.
> - **MBPP-ET** and **HumanEval-ET**: Extended test suites that add 100 hidden test cases per task, offering a more rigorous evaluation of generalization and robustness - helping mitigate overfitting to visible tests.
> - **CodeContests**: A challenging competitive programming benchmark that requires multi-step reasoning and broader generalization.
> - **DS-1000**: A domain-specific benchmark targeting data science tasks. In this dataset, the problem is defined without heavily relying on test cases (one sample is shown as part of the problem prompt).
>
> Notably, the benchmarks HumanEval, CodeContests, DS-1000 contain hidden test cases which help mitigate overfitting to visible tests.
>
> Our method achieves state-of-the-art performance across all major benchmarks: MBPP (96.6%), MBPP-ET (73.0%), HumanEval (99.4%), HumanEval-ET (89.0%), CodeContests (60.6%),  and DS-1000 (69.9%).
> We compared against every strong baseline we could find, both from PapersWithCode and directly from papers. We weren’t just relying on reported results, we actively tried to reproduce methods ourselves wherever possible. In many cases, we reran existing methods on the model (DeepSeek‑V3‑0324), to ensure a fair comparison. When code didn’t work with DeepSeek or wasn’t available, we adapted or re-implemented it, and documented any limitations.
>
> These results show that our method is not limited to MBPP. It generalizes effectively across diverse levels of complexity, from foundational problems to competitive programming tasks and domain-specific benchmarks - demonstrating the broader applicability of our method.
> Due to character limits, we have distributed the result tables across fegN and FLp2.
>
> ## Effect of More Trials on Accuracy
> Thank you for the suggestion. We agree that showing how accuracy scales with increased inference-time compute would provide valuable insight into the efficiency of our method. In our revised version, we will include such a plot.
>
> ## Questions
> > **How does the computational cost (e.g. API-call tokens) of the EG-CFG method compare with other baselines? If the costs differ significantly, the comparison may not be fair.**
>
> First, native parallelization is a key strength of our method. We leverage the low price of existing APIs and the ability to parallel them, which is a growing trend among today's agents. Other baseline methods mentioned in our paper can also have access to full parallel compute, but by design, they cannot fully leverage it. They rely on sequential retries, where each iteration depends on the output of the previous one. This makes it impossible for them to parallelize inference on a single task, even if the computational resources are available.
> Moreover, in (5/6) of the benchmarks: HumanEval, HumanEval-ET, MBPP-ET, CodeContests, DS-1000 there are hidden-test cases. There are hidden test cases. This means our method cannot rely on extensive token usage to overfit the public tests in order to succeed.
> Additionally, In order to rigorously make sure the comparison is fair, we ran trials using the DeepSeek-V3 model with two baseline methods: MGDebugger and MapCoder, on the MBPP benchmark, using a much higher number of retries which leads to significantly more extensive token usage. For MGDebugger, we increased the retry count from 5 to 200 - x40 increase, resulting in ~x40 tokens. This resulted in only an improvement of 34 out of 500 (6.8 percent), out of the 86 examples it originally got wrong. For MapCoder, we similarly increased retries from 5 to 200 (40x tokens), and saw an improvement of just 8 out of 500 (1.6 percent), out of 64 examples it originally failed.
> In both cases, the improvement was achieved with a 40x increase in runtime and is still considerably below our performance on MBPP.
> This supports our conclusion that the advantage of our method is not due to higher token usage, and therefore the comparison is fair. In order to clarify this issue more we will add discussion about this point in our revised version.
>
> > **The MBPP dataset contains relatively simple tasks with few variables. How does the method perform when applied to problems with more variables and inputs? This could increase the complexity of inline execution feedback and potentially result in an overly long prompt.**
>
> Our method is completely agnostic to the number of variables in the input or whether there are variables at all in the input. This is since it is agnostic to the format of the execution feedback (line 196). It relies on implicit supervision, which is an important part of our novelty, as we treat execution feedback as a soft guidance signal. The model interprets execution behavior on its own, without being explicitly told what is correct or relying on any specific format. This allows it to self-verify regardless of the quality of the test cases.
> Moreover, we believe that in many cases, it's actually easier for the model to self-verify logic when it is broken down into smaller, more atomic steps. To encourage this, we introduced the “long-code” prompt template (see Figure 4), which encourages the model to produce longer, step-by-step solutions with simpler lines of code.
> Regarding rigorous evaluation, we have addressed your point in “Significance: Benchmark Diversity”. The benchmarks CodeContests and DS-1000 are indeed more complex, with many more inputs resulting in longer prompts, and our method still outperforms all known results on these benchmarks.
>
> We sincerely appreciate the time and care you put into your review. Your feedback helped us improve the clarity and depth of the paper. If you feel the changes address your concerns, we would be grateful if you consider updating your score. Thank you again for your thoughtful input.

---

> > ### Comment · Reviewer_GpF1 · 2025-08-05
> >
> > I thank the authors for providing additional experimental results to strengthen the significance. The clarifications have addressed most of my concerns. I have one remaining question: for more complex tasks with long inputs, will the method be constrained by the context window length of current LLMs?

---

> > > ### Author Response · Authors · 2025-08-05
> > >
> > > Thank you for the insightful question.
> > > To test possible limits due to the context window size, we generated a random hash function (a meaningful function whose length can be arbitrarily scaled in a controlled manner) and measured the length of its execution trace. Even for functions with 1k lines of code, the trace remained well within the context window of DeepSeek V3. This demonstrates that our method can handle long self-contained functions.
> > >
> > > Moreover, even under extreme context window limitations (for example, when using small models only), our method remains effective. As shown in our ablation study, using a minimal trace still yields strong performance, improving accuracy from 49.4% to 76.4%. A minimal trace captures only the final values and types of all variables in scope after the last line of the solution function has executed, and its size is negligible relative to the LLM's context window.
> > >
> > > Please note that rather than writing a single large block (which is not common practice in software engineering), we suggest seamlessly combining our method with task-decomposition strategies that break problems into multiple simpler, self-contained functions (lines 327-239). This approach aligns with standard software engineering conventions.

---

> > > > ### Comment · Reviewer_GpF1 · 2025-08-07
> > > >
> > > > Thanks to the authors for clarifying that the method remains effective even with very long code (up to 1k lines) and limited context windows. My main concerns have been addressed, so I will slightly raise my score in support.

---

### Official Review · Reviewer_fegN · 2025-06-30

**Clarity:** 4
**Significance:** 3
**Originality:** 3
**Rating:** 4
**Confidence:** 3

**Summary:**

This work proposes a new decoding method for LMs to perform code completion tasks. Unlike prior work, which mainly focuses on having LMs loop between generating code and understanding execution logs at a multi-turn level, this method proposes incorporating execution into the decoding procedure, with LMs being provided execution signals at intermediate points in the code generation process. The authors describe the iterative procedure to extract executable code from intermediate generations, incorporate execution into the in-progress inference output, and use CFG to allow for more dynamic sampling during the decoding process. Evaluating on MBPP, the authors show superior performance over LMs equipped with more traditional execution-feedback techniques, and the ablations back up several of the design decisions made for the decoding pipeline.

**Questions:**

- Line 158 - p_inst is p_0 and p_task combined? Line 170 - where does p_sol fit into this whole process? This is referring to what is eventually regarded as the solution generated by the LM, which is then evaluated for correctness?
- Figure 2 - what is the highlighted “...” character referring to?
- Is there a reason you didn’t compare against agent scaffolds such as SWE-agent, InterCode, CodeAct, or OpenHands? It feels like those works explore the inference-from-execution-across-turns approach that maybe interesting to compare against. At this point, a lot of the code completions benchmarks (e.g. MBPP, HumanEval) feel very saturated, as a strong model like the GPT o* series or Claude 3.7/4 could zero-shot get 90%+ on the benchmark. I feel like the performance gains could be more meaningful if evaluated on a more challenging setting.
- What is “minimal trace” in Table 3?

**Ethical Concerns:**

["NO or VERY MINOR ethics concerns only"]

**Final Justification:**

As I mention in my response to the authors, I maintain my current scores of 4 (rating) and 3 (confidence) in this work.

The authors demonstrate with many more evaluations that this method holds for a broad number of Python code completion benchmarks, which is great to see.

I did not provide a higher score because of the specificity of this method. More complex coding related domains have emerged in the past year (e.g., SWE-bench) and multilingual code completion evaluations are readily available. For a higher score, I would have liked to see this method's generalizability to more languages and domains to justify more significant impact.

**Limitations:**

Yes

**Quality:**

3

**Strengths And Weaknesses:**

Strengths:
* I appreciate the novelty of this work, and I think it is fundamentally quite interesting. Incorporating execution feedback into the decoding process of an LM seems quite neat, and prior works have not really explored this space at all, so it is very neat to see this idea carried out with positive results.
* I liked going through Section 3, and I thought it was written in a very clear and presentable way, where it was very straightforward how each step flowed into the next.
* The results are strong, indicating that this approach works well for MBPP, and more generally, code completion tasks that are written in Python. I think it would be helpful to make the ablations section a bit longer and potentially explain why performance drops can be attributed to certain omissions (e.g. no beam search, CFG), but otherwise the ablations as is very succinctly demonstrate the benefit of each step of the pipeline.

Weaknesses:
* My interpretation is that this approach works exclusively for Python code right now - is this the right understanding? I think this is because of the “Executable Extraction” step, which seems to suggest that the extraction relies heavily on the `ast` library. I’m sure this may be extensible to other languages using the `tree-sitter` library or some non-Python AST construction library, but I feel this adaptation might be non-trivial?
* Since the testbed is mainly function completions (e.g. MBPP, HumanEval), what happens if the code completions are not so “self-contained”? I think this method is quite cool, but I’m not sure how adaptable it is to in-the-wild code completion, as it feels like there’s several criteria that must be satisfied (unit tests, executable env., Python, target completion is for a function) in order for this to work.
* Can you do evaluations with this method on other code completion benchmarks? E.g. HumanEval, LiveCodeBench, BigCodeBench? It’s impressive this method leads to a +14% performance boost (based on Table 1), but I’m curious if this carries over for other code completion settings.

---

> ### Author Rebuttal · Authors · 2025-07-31
>
> We sincerely appreciate your detailed and constructive review. Your positive remarks on the novelty of our approach were encouraging. Your suggestions helped us identify key areas for clarification and refinement of our work. We've made substantial efforts to improve the paper based on your feedback. Below, we respond to each of your comments in detail.
>
> ## Executable Extraction Generality
> EG-CFG is language-agnostic. It only requires the ability to extract executable code segments, execute them, and collect execution feedback. The current implementation is specific to Python and uses the ast library for extracting executable code and a Python debugger that we modified specially for extracting the execution feedback. These can definitely be implemented in other languages.
>
> ## Benchmark Diversity
> We have substantially expanded our evaluation beyond MBPP to address concerns about benchmark diversity and broader applicability.
> Our updated experiments include the additional five benchmarks:
> - **HumanEval**: A foundational and widely adopted benchmark.
> - **MBPP-ET** and **HumanEval-ET**: Extended test suites that add 100 hidden test cases per task, offering a more rigorous evaluation of generalization and robustness - helping mitigate overfitting to visible tests.
> - **CodeContests**: A challenging competitive programming benchmark that requires multi-step reasoning and broader generalization.
> - **DS-1000**: A domain-specific benchmark targeting data science tasks. In this dataset, the problem is defined without heavily relying on test cases (one sample is shown as part of the problem prompt).
>
> Notably, the benchmarks HumanEval, CodeContests, DS-1000 contain hidden test cases which help mitigate overfitting to visible tests.
>
> Our method achieves state-of-the-art performance across all major benchmarks: MBPP (96.6%), MBPP-ET (73.0%), HumanEval (99.4%), HumanEval-ET (89.0%), CodeContests (60.6%),  and DS-1000 (69.9%).
>
> We compared against every strong baseline we could find, both from PapersWithCode and directly from papers. We weren’t just relying on reported results, we actively tried to reproduce methods ourselves wherever possible. In many cases, we reran existing methods on the model (DeepSeek‑V3‑0324), to ensure a fair comparison. When code didn’t work with DeepSeek or wasn’t available, we adapted or re-implemented it, and clearly documented any limitations.
>
> These results show that our method is not limited to MBPP. It generalizes effectively across diverse levels of complexity, from foundational problems to competitive programming tasks and domain-specific benchmarks - demonstrating the broader applicability of our method.
>
> We include detailed benchmark comparison tables below (other tables can be found in reply to FLp2). These present accuracy and retry success rates (RSR) across all datasets and baselines, including both our runs on DeepSeek‑V3‑0324 and results reported in the literature.
>
> **Table**: *Performance on the CodeContests benchmark.*
> Our proposed **EG-CFG** achieves a new state-of-the-art overall accuracy on CodeContests.
> The runs on DeepSeek-V3-0324 are by us, and all other results are quoted from the literature.
> \*LPW results were obtained on a custom test-set; the published code was not compatible with evaluating on DeepSeek.
> \*\*Reported by the MapCoder paper.
> \*\*\*Reported by the CodeSim paper.
> | **Model** | **Method** | **Accuracy (%)** | **RSR (%)** |
> |-----------|------------|------------------|-------------|
> | DeepSeek-V3-0324 | Baseline LLM | 41.81 | 0.00 |
> | DeepSeek-V3-0324 | **EG-CFG (Ours)** | **60.6** | **32.29** |
> | DeepSeek-V3-0324 | MapCoder [Islam et al., 2024] | 50.30 | 14.59 |
> | DeepSeek-V3-0324 | CodeSim [Islam et al., 2025] | 52.72 | 18.76 |
> | GPT-4o | LPW [Lei et al., 2024]* | 34.7 | - |
> | GPT-4o | LDB [Zhong et al., 2024]*** | 29.3 | - |
> | GPT-4 | CodeSim [Islam et al., 2025] | 29.1 | - |
> | GPT-4 | MapCoder [Islam et al., 2024] | 28.5 | - |
> | GPT-4 | Self-Planning [Jiang et al., 2024]** | 10.9 | - |
> | GPT-4 | Analogical [Yasunaga et al., 2023]** | 10.9 | - |
> | GPT-3.5 Turbo | CodeSim [Islam et al., 2025] | 16.4 | - |
> | GPT-3.5 Turbo | MapCoder [Islam et al., 2024] | 12.7 | - |
> | GPT-3.5 Turbo | Analogical [Yasunaga et al., 2023]** | 7.3 | - |
> | GPT-3.5 Turbo | Self-Planning [Jiang et al., 2024]** | 6.1 | - |
> | MoTCoder-15B | MoTCoder [Li et al., 2023] | 26.34 | - |
>
> ## Extending Beyond Self-Contained Functions.
> In order to evaluate our method we used the benchmarks that we mentioned earlier. All of them indeed build upon the assumption of a single entry point “self-contained” function setting.
> As we explicitly wrote in the limitations section (line 326) our method does not exploit a task-decomposition strategy. We believe the core idea of guiding generation via execution feedback can be extended to more complex settings of self contained function and can be combined with task-decomposition and sequential refinements approaches (lines 327-329).
> Regarding the dependency on unit-tests, our method is less dependent on test cases than the existing methods, since in most stages of the generation, it relies on implicit supervision, which is a central part of our method. The execution behavior is used as a soft feedback signal, interpreting outcomes without being explicitly told what is correct. This allows it to self-verify regardless of the quality of the test cases. We explicitly discuss this in the Introduction section (51-55) and highlight it as a key differentiator from prior approaches in the Related Work section.
> We note in the limitation section that the method is applied to benchmarks that have executable test cases. This is the most direct way to compare with the existing relevant methods. However, it is a soft limitation since test cases can also be generated by the model itself from the task instruction.
> We also note that in DS-1000, each problem includes only a single input-output example, making it impractical to rely heavily on such examples during code generation.
>
>
> ## Questions
>
> > **Line 158 - p_inst is p_0 and p_task combined? Line 170 - where does p_sol fit into this whole process? This is referring to what is eventually regarded as the solution generated by the LM, which is then evaluated for correctness?**
>
> Correct. p_0 is a prompt template. We incorporate the task components (text definition of the task, public tests, function name) to the prompt template in-order to form the instruction prompt p_inst. We explicitly explain about the prompt template types in lines 160-163 and include examples of these prompt templates in the appendix A (Figures 3 and 4).
> Thank you for pointing this out - in our revised paper, we will clarify that and make it more explicit.
> p_pre refers to the full model output in response to the instruction prompt p_inst. It includes both the intermediate reasoning and the solution block (which contains the “initial solution”). We define i_sol as the position of the last code-start token (which is defined in p_0) in p_pre, marking the beginning of the solution block. The solution prompt p_sol is constructed by taking the prefix of p_pre up to i_sol and appending the new tokens generated by our decoding algorithm, until a code-end token is generated. The tokens generated beyond i_sol form the final solution returned by our algorithm that is evaluated for correctness.
>
> > **Figure 2 – What is the highlighted “...” character referring to?**
>
> The “...” in Figure 2 indicates parts of the execution trace that were omitted to simplify the figure.
> Below is the complete version of the trace (marked with **)
>
>  Invocation:
>
> first_non_repeating_character("aabc")
>
>  Execution Trace:
>
> s = 'aabc', char_count = {}
>
> char = 'a' -> char_count = {'a': 1}
>
> **
>
> char = 'a' -> char_count = {'a': 2}
>
> char = 'b' -> char_count = {'a': 2, 'b': 1}
>
> char = 'c' -> char_count = {'a': 2, 'b': 1, 'c': 1}
>
> char = 'a' -> count = 2 → skip
>
> char = 'a' -> count = 2 → skip
>
> **
>
> char = 'b' -> count = 1 → return 'b'
>
> > **Is there a reason you didn’t compare against agent scaffolds such as SWE-agent, InterCode, CodeAct, or OpenHands? It feels like those works explore the inference-from-execution-across-turns approach that maybe interesting to compare against. At this point, a lot of the code completions benchmarks (e.g. MBPP, HumanEval) feel very saturated, as a strong model like the GPT o* series or Claude 3.7/4 could zero-shot get 90%+ on the benchmark. I feel like the performance gains could be more meaningful if evaluated in a more challenging setting.**
>
> As described in the “benchmark diversity” we addressed your point regarding evaluation in more diverse and challenging settings of code generation tasks.
> Regarding the benchmark that you proposed, as we understand, these benchmarks (SWE-Agent, InterCode, CodeAct, and OpenHands) are not purely code generation benchmarks in the traditional sense. Rather than focusing solely on evaluating a model’s coding or algorithmic skills, these benchmarks assess broader agentic abilities that reflect task-solving ability in interactive environments, such as interacting with external tools or completing multi-step workflows.
> As we wrote in the conclusion section (lines 310-313), we believe that the core idea of guiding the inference process by the execution feedback can extend to broader settings, which involve combined textual and visual feedback from interactive environments and external programs.
>
> > **What is “minimal trace” in Table 3?**
>
> “Minimal trace” refers to a minimal execution trace that captures only the final values and types of all variables in scope after the last line of the solution function has completed execution. We will extend the current clarification in lines 288-289.
>
> Thanks again for really diving into our paper. We have done our best to improve the work based on your suggestions. If you feel the updates address your concerns, we’d truly appreciate it if you considered updating your score.

---

### Official Review · Reviewer_7xf3 · 2025-07-04

**Clarity:** 1
**Significance:** 3
**Originality:** 3
**Rating:** 5
**Confidence:** 3

**Summary:**

The authors present a method for generating solutions to the MBPP (mostly Basic Python Programs) dataset by forming incomplete candidate programs consisting of a small number of lines, then iteratively: executing them (after some minor syntactic repair to make them executable) while tracing the internal state, and prompting the model to further continue the programs by appending more lines to the end. Compared to other iterative generation approaches, which largely rely on debugging and editing already-completed programs, the approach focuses on executing unfinished programs and using the execution information to further extend them. They compare against several other methods and report state-of-the-art results on the MBPP dataset.

**Questions:**

Questions:
- How was beam search performed with lines of code as the units for the length of the samples, since a fixed number of lines of code doens't translate into a fixed number of tokens?
- In equation 10, what happens if some value of $\gamma$ causes $\log M_{CFG}(\cdots)$ to become a positive number? I think this can happen unless $0 \le \gamma \le 1$.
- When does the loop end if a correct solution is not found (i.e. how long do you try before you give up)?
- What happens to the initial solution $w^0_{pre}, \cdots, w_
- What is the relationship between $p_{sol}$ and $p_{pre}$?

I found it hard to follow exactly the procedure used and would have appreciated 1) more examples of the prompts used with the models, 2) a diagram explaining the loop, 3) the overall procedure expressed as pseudocode.

Line 168: "last few example inside the template" should be "last few-shot example inside the template"?

**Ethical Concerns:**

["NO or VERY MINOR ethics concerns only"]

**Final Justification:**

The authors addressed significant concerns in the paper regarding experimental results, which I am putting high weight on my decision to raise the score from 3 to 5. The clarity issues are hard to resolve through the rebuttal given its format, and the authors did not include the specific new language they would use in a revised paper, but I think those can be considered more minor.

**Limitations:**

Yes

**Paper Formatting Concerns:**

No concerns with formatting

**Quality:**

3

**Strengths And Weaknesses:**

## Strengths
- The paper reports SoTA results on a dataset that many others have attempted.
- The method tries out an interesting combination of classifier-free guidance with iterative generation of candidate programs while providing execution feedback.

## Weaknesses
- I found the procedure in the paper a bit hard to understand, for example:
  - Some parts (like equation 3) discuss steps as tokens during decoding; but in other parts, there is a different notion of a step which occurs due to gradual extension of canddiate solutions, which isn't as clearly described.
  - There is an outer loop for extending solutions but it is not explicitly and clearly stated how it works, e.g. how many times it would run and what are the termination conditions.
  - The details of the execution traces is not clearly stated.
- The experiments are only on MBPP, which is now a few years old and superceded by other similar datasets such as LiveCodeBench. - - As I understand it, MBPP has no hidden test cases, which reduces the realism of the benchmark; the procedure outlined might have overfit to that aspect of the benchmark, considering aspects like "we return the first correct solution identified".
- Because of the lack of hidden test cases, a method that tries all possible programs (given infinite time) would get 100% accuracy on MBPP. It wasn't clear to me whether the comparison to baseline methods is fair in the sense that this method might have used significantly more computation than the alternatives, or significantly more trials. Only wall-clock time with "full parallelization" is reported as a comparison.

I would have trouble recommending the paper for acceptance until at least the clarity-related concerns are resolved, and I would suggest trying the approach on other datasets than MBPP as well.

(REVISION AFTER REBUTTAL): While the clarity-related concerns are hard to evaluate, I believe the authors have significantly addressed my concerns about generalizability and inference budget.

---

> ### Author Rebuttal · Authors · 2025-07-31
>
> Thank you for your thoughtful and detailed review. Your comments helped us identify several areas that we want to clarify and we have worked hard to improve the paper accordingly. Below, we address each of your points in turn.
>
> ## Clarity Related Concerns
> > Ambiguity Between Token Steps and Candidate Extensions
>
> On the one hand, we used token notation in Eq. (3) and (7), as these parts are related to token sampling and indexing of specific tokens. On the other hand, we chose to use the granularity of code snippets in the candidate continuation sampling in Eq. (4), because after sampling candidate continuation programs we perform deduplication and execution which are both operating at the granularity of full code snippets rather than individual tokens. For consistency and extra clarity, we will revise Eq. (4) to use token notation as well.
>
> > Outer Loop Behavior and Termination Conditions
>
> The outer loop is the execution of each agent as an independent inference run, which terminates (returns) when the first correct solution is found. Otherwise, inference continues across the remaining agents (lines 229 to 231). Each inference run is independent and allows parallel execution (line 274). We did not describe a retry mechanism because there is no such mechanism, i.e., each agent runs exactly once. To make this clearer, we will revise the Method section to describe the outer loop execution as such and will include corresponding pseudocode.
>
> > Execution Trace Format Was Unclear
>
> Our approach is agnostic to the format of the execution feedback (line 196). This is an important part of our novelty, as we treat execution feedback as a soft guidance signal. The model interprets execution behavior on its own, without being explicitly told what is correct or relying on any specific format (lines 141–142). The structure of the execution trace used in our implementation is described in lines 197–200. To further illustrate this, we included Figure 2, which shows a concrete example of a trace. In our revision, we will formalize the trace structure so that the exact format is fully explicit.
>
> > Clarifying the Inference Procedure
>
> In our revision, we will include full pseudocode outlining the complete procedure. We will split it into two algorithms: Algorithm 1 describes the outer loop that executes agents in parallel, each executing our core algorithm “Execution-Guided Inference Loop” (Algorithm 2). In addition, we will add more prompt examples and a diagram.
>
> ## Benchmark Diversity
> We have substantially expanded our evaluation beyond MBPP to address concerns about benchmark diversity and broader applicability.
> Our updated experiments include the additional five benchmarks:
> - **HumanEval**: A foundational and widely adopted benchmark.
> - **MBPP-ET** and **HumanEval-ET**: Extended test suites that add 100 hidden test cases per task, offering a more rigorous evaluation of generalization and robustness - helping mitigate overfitting to visible tests.
> - **CodeContests**: A challenging competitive programming benchmark that requires multi-step reasoning and broader generalization.
> - **DS-1000**: A domain-specific benchmark targeting data science tasks. In this dataset, the problem is defined without heavily relying on test cases (one sample is shown as part of the problem prompt).
>
> Notably, the benchmarks HumanEval, CodeContests, DS-1000 contain hidden test cases which help mitigate overfitting to visible tests.
>
> Our method achieves state-of-the-art performance across all major benchmarks: MBPP (96.6%), MBPP-ET (73.0%), HumanEval (99.4%), HumanEval-ET (89.0%), CodeContests (60.6%),  and DS-1000 (69.9%).
> We compared against every strong baseline we could find, both from PapersWithCode and directly from papers. We weren’t just relying on reported results, we actively tried to reproduce methods ourselves wherever possible. In many cases, we reran existing methods on the model (DeepSeek‑V3‑0324), to ensure a fair comparison. When code didn’t work with DeepSeek or wasn’t available, we adapted or re-implemented it, and clearly documented any limitations.
>
> These results show that our method is not limited to MBPP. It generalizes effectively across diverse levels of complexity, from foundational problems to competitive programming tasks and domain-specific benchmarks - demonstrating the broader applicability of our method.
>
> We include detailed benchmark comparison tables below (other tables can be found in reply to fegN and FLp2). These present accuracy and retry success rates (RSR) across all datasets and baselines, including both our runs on DeepSeek‑V3‑0324 and results reported in the literature.
>
>
> ## ​​Fairness of Computational Comparison
> First, native parallelization is a key strength of our method. We leverage the low price of existing APIs and the ability to parallel them, which is a growing trend of today's agents. Other baseline methods mentioned in our paper can also have access to full parallel compute, but by design, they cannot fully leverage it. They rely on sequential retries, where each iteration depends on the output of the previous one. This makes it impossible for them to parallelize inference on a single task, even if the computational resources are available.
> Moreover, in (5/6) of the benchmarks: HumanEval, HumanEval-ET, MBPP-ET, CodeContests, DS-1000 there are hidden-test cases. There are hidden test cases. This means our method cannot rely on extensive token usage to overfit the public tests in order to succeed.
>
> Additionally, In order to rigorously make sure the comparison is fair, we ran trials using the DeepSeek-V3 model with two baseline methods: MGDebugger and MapCoder, on the MBPP benchmark, using a much higher number of retries which leads to significantly more extensive token usage. For MGDebugger, we increased the retry count from 5 to 200 - x40 increase, resulting in ~x40 tokens. This resulted in only an improvement of 34 out of 500 (6.8 percent), out of the 86 examples it originally got wrong. For MapCoder, we similarly increased retries from 5 to 200 (40x tokens), and saw an improvement of just 8 out of 500 (1.6 percent), out of 64 examples it originally failed.
> In both cases, the improvement was achieved with a 40x increase in runtime and is still considerably below our performance on MBPP.
>
> This supports our conclusion that the advantage of our method is not due to higher token usage, and therefore the comparison is fair. In order to clarify this issue more we will add discussion about this point in our revised version.
>
> ## Questions
>
> > **How was beam search performed with lines of code as the units for the length of the samples, since a fixed number of lines of code doens't translate into a fixed number of tokens?**
>
> Sampling is performed auto-regressively with a stopping condition based on the number of newline characters (\n) encountered. Specifically, generation stops once the target number of lines, denoted by d, is reached. We do not assume a fixed number of tokens per line. We explicitly wrote in lines 175-176.
>
> > **In equation 10, what happens if some value of  gamma causes M_CFG to become a positive number? I think this can happen unless.**
>
> Thank you for clarifying this point. For some values of gamma, the guided logits M_CFG​ can include positive values and won’t sum to a valid probability distribution. When using argmax decoding rather than sampling, normalization isn’t necessary. Softmax wouldn’t change the selected token, so the behavior of the algorithm remains the same. However, it is possible to normalize the distribution to make it compatible with the standard inference loop that allows sampling.
>
> > **When does the loop end if a correct solution is not found (i.e. how long do you try before you give up)?**
>
> We do not use a retry mechanism within a single agent. Instead, we run multiple agents in parallel, each following a different reasoning path by varying dynamic signal parameters. As soon as one agent finds a correct solution, the others are terminated, and that solution is returned. If none succeed, no solution is reported for that task - “not found”. We can add a retry mechanism, which was shown to be beneficial to iterative refinement methods.
>
> > **What happens to the initial solution $w^0_{pre}, \cdots, w_**
> > **What is the relationship between p_sol  and p_pre ?**
>
> p_pre refers to the full model output in response to the instruction prompt p_inst. It includes both the intermediate reasoning and the solution block (which contains the “initial solution”). We define i_sol as the position of the last code-start token in p_pre, marking the beginning of the solution block. The solution prompt p_sol is constructed by taking the prefix of p_pre up to i_sol (We ignore the tokens representing “initial solution” which come after that prefix) by appending the new tokens generated by our decoding algorithm, until a code-end token is generated. The tokens generated beyond i_sol form the final solution returned by our algorithm.
>
> > **Line 168: "last few example inside the template" should be "last few-shot example inside the template"?**
>
> Correct - fixed.
>
> In conclusion, we sincerely thank you for your thoughtful and detailed review. We've worked hard to address each of your concerns through significant clarifications, expanded experiments, and improved presentation of our method. We hope these revisions make the contribution and impact of our work clearer. If you find the updated version compelling, we kindly ask you to consider updating your score. We truly appreciate your time and feedback.

---

> ### Author Response · Authors · 2025-08-06
>
> Thank you again for taking the time to review our paper.
>
> We kindly ask for your response to our new comments before the end of the rebuttal period.
>
> We have worked extensively to address your primary concerns, including clearing various aspects of our method and massively extending our evaluation beyond the MBPP dataset.
>
> Please let us know if our responses address your concerns or if we can clarify any remaining issues. Thank you once again for your time and consideration.

---

### Official Review · Reviewer_FLp2 · 2025-07-05

**Clarity:** 3
**Significance:** 2
**Originality:** 3
**Rating:** 4
**Confidence:** 4

**Summary:**

This paper introduces a novel code generation method called Execution-Guided Classifier-Free Guidance. The core idea is to incorporate real-time execution feedback during the inference process of a language model, mimicking how human programmers iteratively run partial code to guide development.

**Questions:**

Refer to the weaknesses please.

**Ethical Concerns:**

["NO or VERY MINOR ethics concerns only"]

**Final Justification:**

My concern is well addressed, so I tend to vote for the acceptance of this paper.

**Limitations:**

Refer to the weaknesses please.

**Quality:**

3

**Strengths And Weaknesses:**

Strengths:
1. The paper is well-writing and easy to follow.
2. The proposed method is inference-based without introducing the additional training cost.
3. Simple, single-agent setup makes the method easy to understand and reproduce.
4. The performance for the evaluation on MBPP is good.

Weaknesses:
1. The evaluation is not diversity enough. It may be possible that the proposed method's generalization ability is not so clear. Is it possible to extend to some more challenging benchmark like LivecodeBench, and other coding tasks like DS-1000 (data science) and Bird (SQL)?
2. Relies on test cases for feedback, which may limit applicability to real-world tasks lacking such supervision. Or would the methods highly rely on the high quality and reliable test cases? At least, the paper should discuss more about corresponding impact and possible limitations.
3. Introduces significant computational overhead due to repeated execution and beam search. Would it be possible to discuss the inference time and token number compared with other baselines?

---

> ### Author Rebuttal · Authors · 2025-07-31
>
> Thank you for your thoughtful review and constructive feedback.
>
> ## Evaluation Diversity and Generalization
> We have substantially expanded our evaluation beyond MBPP to address concerns about benchmark diversity and broader applicability.
> Our updated experiments include the additional five benchmarks:
> - **HumanEval**: A foundational and widely adopted benchmark.
> - **MBPP-ET** and **HumanEval-ET**: Extended test suites that add 100 hidden test cases per task, offering a more rigorous evaluation of generalization and robustness - helping mitigate overfitting to visible tests.
> - **CodeContests**: A challenging competitive programming benchmark that requires multi-step reasoning and broader generalization.
> - **DS-1000**: A domain-specific benchmark targeting data science tasks. In this dataset, the problem is defined without heavily relying on test cases (one sample is shown as part of the problem prompt).
>
> Notably, the benchmarks HumanEval, CodeContests, DS-1000 contain hidden test cases which help mitigate overfitting to visible tests.
>
> Our method achieves state-of-the-art performance across all major benchmarks: MBPP (96.6%), MBPP-ET (73.0%), HumanEval (99.4%), HumanEval-ET (89.0%), CodeContests (60.6%), CodeContests (60.6%) and DS-1000 (69.9%).
>
> We compared against every strong baseline we could find, both from PapersWithCode and directly from papers. We weren’t just relying on reported results, we actively tried to reproduce methods ourselves wherever possible. In many cases, we reran existing methods on the model (DeepSeek‑V3‑0324), to ensure a fair comparison. When code didn’t work with DeepSeek or wasn’t available, we adapted or re-implemented it, and clearly documented any limitations.
> These results show that our method is not limited to MBPP. It generalizes effectively across diverse levels of complexity, from foundational problems to competitive programming tasks and domain-specific benchmarks - demonstrating the broader applicability of our method.
>
> We include detailed benchmark comparison tables below (due to character limit, CodeContests table can be found in reply to fegN). These present accuracy and retry success rates (RSR) across all datasets and baselines, including both our runs on DeepSeek‑V3‑0324 and results reported in the literature.
>
> **Table**: *Performance on the MBPP and MBPP-ET benchmarks.*
> Our proposed **EG-CFG** achieves a new state-of-the-art overall accuracy.
> The DeepSeek–Coder-1.3B and –V3-0324 results for all baselines were obtained by our study using the official implementations provided by each baseline method.
> The results below the double separator were collected from the respective papers.
>
> | **Model** | **Method** | **MBPP Acc. (%)** | **MBPP RSR (%)** | **MBPP-ET Acc. (%)** | **MBPP-ET RSR (%)** |
> |-----------|------------|-------------------|------------------|----------------------|---------------------|
> | DeepSeek-Coder 1.3B | Baseline LLM | 49.4 | 0.0 | 42.6 | 0.0 |
> | **DeepSeek-Coder 1.3B** | **EG-CFG (Ours)** | 83.2 | 66.79 | 59.8 | 29.96 |
> | DeepSeek-Coder 1.3B | MapCoder [Islam et al., 2024] | 55.2 | 11.46 | 46.2 | 6.27 |
> | DeepSeek-Coder 1.3B | MGDebugger [Shi et al., 2024] | 70.4 | 41.5 | 44.6 | 3.48 |
> | DeepSeek-V3-0324 | Baseline LLM | 82.8 | 0.0 | 64.8 | 0.0 |
> | **DeepSeek-V3-0324** | **EG-CFG (Ours)** | **96.6** | **80.23** | **73.0** | **23.29** |
> | DeepSeek-V3-0324 | MapCoder [Islam et al., 2024] | 87.2 | 25.58 | 69.6 | 13.63 |
> | DeepSeek-V3-0324 | MGDebugger [Shi et al., 2024] | 86.8 | 23.25 | 64.8 | 0.0 |
> | DeepSeek-V3-0324 | LPW [Lei et al., 2024] | 84.0 | 6.97 | 65.2 | 1.13 |
> | GPT-4 | Baseline LLM | 68.3 | - | 49.2 | - |
> | GPT-4 | Self-Collaboration [Self-Collaboration] | 78.9 | - | 62.1 | - |
> | GPT-4 | Self-Debugging [Chen et al., 2023] | 80.6 | - | - | - |
> | GPT-4 | MetaGPT [MetaGPT] | 87.7 | - | - | - |
> | GPT-4 | MapCoder [Islam et al., 2024] | 83.1 | - | 57.5 | - |
> | GPT-4o | LPW [Lei et al., 2024] | 84.8 | - | 65.8 | - |
> | CodeQwen1.5 | MGDebugger [Shi et al., 2024] | 80.8 | - | - | - |
> | DeepSeek-Coder-V2-Lite | MGDebugger [Shi et al., 2024] | 80.0 | - | - | - |
> | Claude-Sonnet-3.5 | Baseline LLM [Hu et al., 2025] | 88.7 | - | - | - |
> | Claude-Sonnet-3.5 | QualityFlow [Hu et al., 2025] | 94.2 | - | - | - |
>
> **Table**: *Performance on the HumanEval and HumanEval-ET benchmarks.*
> Our proposed **EG-CFG** achieves a new state-of-the-art overall accuracy on HumanEval-ET.
>
> | **Model** | **Method** | **HumanEval Acc. (%)** | **HumanEval RSR (%)** | **HumanEval-ET Acc. (%)** | **HumanEval-ET RSR (%)** |
> |-----------|------------|------------------------|-----------------------|---------------------------|--------------------------|
> | DeepSeek-V3-0324 | Baseline LLM | 82.92 | 0.0 | 79.20 | 0.0 |
> | **DeepSeek-V3-0324** | **EG-CFG (Ours)** | **99.4** | **94.04** | **89.02** | **47.21** |
> | DeepSeek-V3-0324 | MapCoder [Islam et al., 2024] | 96.95 | 82.14 | 81.70 | 12.02 |
> | DeepSeek-V3-0324 | MGDebugger [Shi et al., 2024] | 87.20 | 25.05 | 81.09 | 9.09 |
> | DeepSeek-V3-0324 | LPW [Lei et al., 2024] | 95.12 | 71.42 | 84.74 | 26.63 |
> | GPT-4 | Baseline LLM | 67.7 | - | 50.6 | - |
> | GPT-4 | Self-Collaboration [Self-Collaboration] | 90.7 | - | 70.1 | - |
> | GPT-4 | Self-Debugging [Chen et al., 2023] | 61.6 | - | 45.8 | - |
> | GPT-4 | MetaGPT [MetaGPT] | 85.9 | - | - | - |
> | GPT-4 | MapCoder [Islam et al., 2024] | 80.5 | - | 70.1 | - |
> | GPT-4o | LPW [Lei et al., 2024] | 98.2 | - | 84.8 | - |
> | LLaMA 3 | LDB [Zhong et al., 2024] | 99.4 | - | - | - |
> | Claude-Sonnet-3.5 | QualityFlow [Hu et al., 2025] | 98.8 | - | - | - |
> | CodeQwen1.5 | MGDebugger [Shi et al., 2024] | 91.5 | 64.1 | - | - |
> | DeepSeek-Coder-V2-Lite | MGDebugger [Shi et al., 2024] | 94.5 | 76.3 | - | - |
>
> **Table**: *Accuracy results from the official DS-1000 leaderboard.*
> | **Model** | **Method** | **Accuracy** | **RSR** |
> |-----------|------------|--------------|---------|
> | **DeepSeek-V3-0324** | **EG-CFG (Ours)** | **69.9%** | **50.73%** |
> | DeepSeek-V3-0324 | Baseline LLM | 38.9% | 0.00% |
> | gpt-4o-2024-08-06 | Baseline LLM | 59.9% | - |
> | claude-3-5-sonnet-20240620 | Baseline LLM | 54.3% | - |
> | gpt-4-turbo-2024-04-09 | Baseline LLM | 54.0% | - |
> | deepseek-ai-deepseek-coder-V2-SFT | Baseline LLM | 53.2% | - |
> | Qwen-Qwen2-72B-Instruct | Baseline LLM | 52.8% | - |
> | deepseek-chat-V2.5 | Baseline LLM | 51.2% | - |
> | mistralai-Codestral-22B-v0.1 | Baseline LLM | 51.2% | - |
> | gpt-4-0613 | Baseline LLM | 51.0% | - |
> | gpt-4o-mini-2024-07-18 | Baseline LLM | 50.5% | - |
>
>
> ## Dependence on Test Cases
> Our method is less dependent on test cases than the existing methods, since in most stages of the generation, it relies on implicit supervision, which is a central part of our method. The execution behavior is used as a soft feedback signal, interpreting outcomes without being explicitly told what is correct. This allows it to self-verify regardless of the quality of the test cases. We explicitly discuss this in the Introduction section (51-55) and highlight it as a key differentiator from prior approaches in the Related Work section (lines 141-143).
> We note in the limitation section that the method is applied to benchmarks that have executable test cases. This is the most direct way to compare with the existing relevant methods. However, it is a soft limitation since test cases can also be generated by the model itself from the task instruction.
> Finally, in five out of six benchmarks we evaluate on (HumanEval, HumanEval-ET, MBPP-ET, CodeContests, and DS-1000), the test cases are hidden. This means the model never sees them during inference. It helps ensure that our method’s strong performance is not due to overfitting to public test cases and further supports its generalization capabilities.
> We also note that in DS-1000, each problem includes only a single input-output example, making it impractical to rely heavily on such examples during code generation.
>
> ## Computational Overhead
> First, native parallelization is a key strength of our method. We leverage the low price of existing APIs and the ability to parallel them, which is a growing trend of today's agents. Other baseline methods mentioned in our paper can also have access to full parallel compute, but by design, they cannot fully leverage it. They rely on sequential retries, where each iteration depends on the output of the previous one. This makes it impossible for them to parallelize inference on a single task, even if the computational resources are available.
> Moreover, in (5/6) of the benchmarks: HumanEval, HumanEval-ET, MBPP-ET, CodeContests, DS-1000 there are hidden-test cases. There are hidden test cases. This means our method cannot rely on extensive token usage to overfit the public tests in order to succeed.
> Additionally, In order to rigorously make sure the comparison is fair, we ran trials using the DeepSeek-V3 model with two baseline methods: MGDebugger and MapCoder, on the MBPP benchmark, using a much higher number of retries which leads to significantly more extensive token usage. For MGDebugger, we increased the retry count from 5 to 200 - x40 increase, resulting in ~x40 tokens. This resulted in only an improvement of 34 out of 500 (6.8 percent), out of the 86 examples it originally got wrong. For MapCoder, we similarly increased retries from 5 to 200 (40x tokens), and saw an improvement of just 8 out of 500 (1.6 percent), out of 64 examples it originally failed.
> In both cases, the improvement was achieved with a 40x increase in runtime and is still considerably below our performance on MBPP.
> This supports our conclusion that the advantage of our method is not due to higher token usage, and therefore the comparison is fair. In order to clarify this issue more we will add discussion about this point in our revised version.

---

> ### Author Response · Authors · 2025-08-06
>
> Thank you once again for taking the time to review our paper.
>
> Following your feedback, we have worked diligently to address your primary concerns, and in particular, present a much more diverse body of empirical results.
>
> We would greatly appreciate it if you could respond to our comments before the end of the rebuttal period so that we could address any remaining issues.
>
> Thank you for your time and consideration.

---

### Comment · Area_Chair_qym4 · 2025-08-05
**Kind remind for author-reviewer discussion**

Dear Authors and Reviewers,

Thank you for submitting and reviewing the papers to contribute to the conference. This is a kind remind that the due date of author-reviewer discussion is coming soon. Please participate the discussion to clarify paper statement or concerns.

Thanks!

AC

---

### Author Response · Authors · 2025-08-08
**Non-Python Evaluation: CUDA Code Optimization via EG-CFG**

During the rebuttal period, we were able to show results for five additional important code generation benchmarks, achieving state-of-the-art results, using open-weights models, compared to all previously published results, including the latest closed models.

All the results we present are for Python benchmarks. Reviewers fegN and FLp2 have wondered about other programming languages. Applying our method to a multilingual benchmark involves substantial infrastructure work for each language, performing tasks such as extracting executable code segments from LLM outputs, compiling and executing the code, and adapting a debugger or tracing tool to collect execution traces.

Motivated by the reviews, and to move beyond standard Python code generation and showcase the diversity and broader applicability of our approach, we explored a novel and practically relevant task: generating CUDA kernel function in C that is functionally equivalent to a given PyTorch function but optimized for GPU performance. In this setting, input-output pairs are readily available, but writing efficient CUDA code demands significant domain expertise.

## Dataset
To evaluate our method on diverse CUDA optimization scenarios, we asked ChatGPT-4o to generate eight synthetic PyTorch functions that are intentionally suboptimal but well-suited for GPU-level optimization. Each function has the same signature: it takes two input tensors a and b of shape (N,) and returns an output tensor of the same shape. The generated functions are designed to reflect common inefficiencies found in real-world code, such as warp divergence, loop-carried dependencies, redundant computations, uncoalesced memory access, and suboptimal arithmetic operations. These artifacts make the functions functionally correct in PyTorch, but perform poorly when executed on a GPU. As such, they present meaningful opportunities for transformation into highly optimized CUDA kernels.

The prompt we used to generate these samples was:
“Please create eight examples of simple PyTorch functions that can be optimized when compiled to CUDA. Each function should receive two 1D tensors a and b and return a single output tensor of the same length. Each example should contain a known inefficiency (e.g., warp divergence, redundant computation, poor memory access) that could be improved with CUDA-level optimization.”

## Setup
All experiments were conducted on an AWS EC2 g4dn.xlarge instance running Ubuntu 22.04 equipped with a single NVIDIA Tesla T4 GPU (15 GB GDDR6 memory). The system was running CUDA 12.8 and NVIDIA driver version 570.172.08.

## Metrics

We evaluate GPU performance of the generated CUDA kernel functions using NVIDIA's profiling tool Nsight Compute (NCU), with input tensors of size N = 256 populated with random values generated using a fixed seed. The following hardware-level metrics are collected: executed IPC, warp cycles per instruction, SM utilization, occupancy, L1/L2 cache hit rates, and branch efficiency.

---

> ### Author Response · Authors · 2025-08-08
> **Non-Python Evaluation: CUDA Code Optimization via EG-CFG (part 2)**
>
> ## Metrics
>
> We evaluate GPU performance of the generated CUDA kernel functions using NVIDIA's profiling tool Nsight Compute (NCU), with input tensors of size N = 256 populated with random values generated using a fixed seed. The following hardware-level metrics are collected: executed IPC, warp cycles per instruction, SM utilization, occupancy, L1/L2 cache hit rates, and branch efficiency.
>
> ### Formulation
>
> All metrics are normalized using **z-score normalization**:
>
> $$
> x_i^{\text{norm}} = \frac{x_i - \mu_i}{\sigma_i}
> $$
>
> where \$\mu\_i\$ and \$\sigma\_i\$ are the global mean and standard deviation of metric \$i\$, computed across all samples (baseline LLM and our method combined so that the normalization is shared).
> If \$\sigma\_i = 0\$, we define \$x\_i^{\text{norm}} = 0\$.
>
> A scalar **performance score** is then computed as a **weighted sum** over the normalized metrics:
>
> $$
> \text{score} = \sum_i w_i \cdot x_i^{\text{norm}}
> $$
>
> Lower scores indicate better GPU performance.
>
> To obtain a single score, we propose the `normalized_weighted_sum` variant, which uses the following weights (**lower score = better**):
>
> ### Metric Weights Used in Normalized Scoring
>
> | Metric                  | Role in Performance                       | Weight | Sign Explanation                          | Norm Explanation (Significance)                     |
> | ----------------------- | ----------------------------------------- | ------ | ----------------------------------------- | --------------------------------------------------- |
> | `executed_ipc`          | Instruction throughput (higher is better) | +1.00  | Higher IPC = better parallelism           | Most critical metric - directly reflects throughput |
> | `warp_cycles_per_instr` | Cycles per instruction (lower is better)  | −1.00  | Lower latency = better performance        | Inversely related to IPC - equally important        |
> | `sm_busy_pct`           | SM (core) utilization                     | +0.50  | Higher SM activity = more work done       | Important, but less direct than IPC                 |
> | `occupancy_pct`         | Occupancy of SM slots                     | +0.50  | Higher occupancy = more threads in flight | Indicates scheduling efficiency                     |
> | `l1_hit_rate`           | L1 cache hit rate                         | +0.20  | Higher = better memory locality           | Helpful, but secondary optimization                 |
> | `l2_hit_rate`           | L2 cache hit rate                         | +0.20  | Higher = better global memory efficiency  | Improves bandwidth use                              |
> | `branch_efficiency_pct` | Branch coherence within warps             | +0.10  | Higher = less warp divergence             | Least critical, but still relevant to performance   |
>
>
> ### Notes:
> * **Positive signs** indicate metrics where *higher is better*.
> * **Negative signs** indicate metrics where *lower is better*.
> * **Norms (magnitudes)** are tuned to balance importance - IPC and warp latency dominate, while memory and branching metrics are secondary.
>
>
> ## Results
> The tables below present performance results comparing EG-CFG to the Baseline LLM across all evaluated tasks. Both methods use the DeepSeek-V3-0324 LLM. The baseline LLM was run multiple times with a temperature of 0.6 to create a diverse set of solutions. For broader comparison, we also evaluated CUDA code generated by an additional frontier model: Gemini Pro 2.5 with its default settings and the same prompt used in the baseline LLM.
> Scores are computed using a normalized weighted sum over profiling metrics. Each metric is normalized using z-score normalization, as described in the Metrics section. Lower performance scores indicate better GPU efficiency.

---

> ### Author Response · Authors · 2025-08-08
> **Non-Python Evaluation: CUDA Code Optimization via EG-CFG (part 3)**
>
> The first table lists the mean performance (z-values, lower is better) per metric for each method.
>
> | Metric                     | Base LLM | EG-CFG   | Gemini 2.5 Pro |
> |---------------------------|----------|----------|----------------|
> | Performance Score         | -0.02    | -0.32 | -0.035         |
> | Executed IPC              | -0.00| -0.00 | -0.0225        |
> | Warp Cycles / Instruction | -0.10| 0.02     | 0.0625         |
> | SM Busy [%]               | 0.05     | 0.04 | -0.0775        |
> | Occupancy [%]             | -0.24    |-0.27| 0.34875        |
> | L1 Hit Rate [%]           | 0.22     | 0.10 | -0.35          |
> | L2 Hit Rate [%]           | -0.35    | -1.06| -0.25875       |
> | Branch Efficiency [%]     | 0.09 | 0.09 | 0.32375        |
>
>
>
> The second table shows the weighted score per sample, comparing the EG-CFG method against the baseline LLM.
>
>
> | Task ID    | BaselineLLM Score | EG-CFG Score | Δ (Abs) | Δ (%)   | Best Score   |
> |------------|-------------------|--------------|---------|---------|----------|
> | example_1  | -0.765            | -1.353       | +0.589  | +77.0%  | EG-CFG   |
> | example_2  | -0.690            | -1.166       | +0.476  | +68.9%  | EG-CFG   |
> | example_3  | 6.332             | 6.170        | +0.162  | +2.6%   | EG-CFG   |
> | example_4  | -1.399            | -1.825       | +0.426  | +30.5%  | EG-CFG   |
> | example_5  | -1.211            | -1.249       | +0.038  | +3.2%   | EG-CFG   |
> | example_6  | -0.697            | -1.105       | +0.408  | +58.5%  | EG-CFG   |
> | example_7  | -0.553            | -0.780       | +0.227  | +41.0%  | EG-CFG   |
> | example_8  | -1.168            | -1.277       | +0.108  | +9.3%   | EG-CFG   |
> | **Mean Δ%**|                   |              |         | **36.4%** |          |
>
>
> The third table compares the weighted performance scores of EG-CFG against Gemini 2.5 Pro.
> | Task ID     | Gemini 2.5 Pro Score | EG-CFG Score | Δ (Abs) | Δ (%)     | Best Score |
> | ----------- | ---------------- | ------------ | ------- | --------- | ---------- |
> | example\_1  | -1.060           | -1.353       | +0.290  | +27.4%    | EG-CFG     |
> | example\_2  | -0.830           | -1.170       | +0.340  | +41.0%    | EG-CFG     |
> | example\_3  | 6.190            | 6.170        | +0.020  | +0.3%     | Gemini 2.5 Pro    |
> | example\_4  | -1.150           | -1.825       | +0.675  | +58.7%    | EG-CFG     |
> | example\_5  | -1.050           | -1.249       | +0.199  | +18.9%    | EG-CFG     |
> | example\_6  | -0.750           | -1.105       | +0.355  | +47.3%    | EG-CFG     |
> | example\_7  | -0.570           | -0.780       | +0.210  | +36.8%    | EG-CFG     |
> | example\_8  | -1.060           | -1.277       | +0.217  | +20.5%    | EG-CFG     |
> | **Mean Δ%** |                  |              |         | **31.4%** |            |
>
> EG-CFG consistently outperforms the baseline LLM across all CUDA code generation tasks, achieving a significantly lower overall performance score (−0.32 vs −0.02) under the normalized weighted scheme. This gain reflects improvements in key hardware-level metrics such as L2 and L1 cache hit rates, occupancy, and SM utilization, leading to more efficient memory access and thread scheduling. While warp cycles per instruction are slightly worse (0.02 vs −0.10), IPC remains equivalent, and the overall efficiency is clearly improved.
>
> Per-task results show EG-CFG outperforming the baseline on all eight examples, with gains ranging from +2.6% to +77.0% and a +36.4% average improvement. These results, achieved with the same underlying LLM, highlight the impact of execution guidance in enabling hardware-aware optimizations beyond the reach of standard prompting.
>
> Compared to Gemini 2.5 Pro, EG-CFG achieves better performance in 7 of 8 tasks, with a +31.4% average improvement-despite using an open-weight model rather than a proprietary one. The first table confirms EG-CFG’s advantage across key metrics, often achieving the best or near-best values.
>
> Altogether, these results demonstrate EG-CFG’s ability to reliably produce more efficient CUDA code than both a naive baseline and a commercial LLM, without retraining or specialized infrastructure. This affirms its practicality and generality for real-world performance-critical code generation.

---

> > ### Author Response · Authors · 2025-08-08
> > **Non-Python Evaluation: CUDA Code Optimization via EG-CFG (part 4 -- first qualitative example)**
> >
> > ## Qualitative Results
> > ## Example 1
> > To illustrate the impact of our method, we present a representative case study (the first example in the benchmark)
> >
> > ### PyTorch Function (Suboptimal)
> >
> > ```python
> > def solution(a: torch.Tensor, b: torch.Tensor) -> torch.Tensor:
> >     out = torch.zeros_like(a)
> >     for i in range(a.shape[0]):
> >         if a[i] > b[i]:
> >             out[i] = a[i] ** 2
> >         else:
> >             out[i] = b[i] ** 2
> >     return out
> > ```
> >
> > **Issue**: This function introduces **warp divergence** due to the per-element conditional (`if a[i] > b[i]`). When threads within a warp follow different branches, the GPU serializes the execution paths, leading to underutilization of SIMD cores.
> >
> > ### Baseline LLM Generated CUDA
> >
> > ```cpp
> > __global__ void solution(const float* a, const float* b, float* out, int N) {
> >     int idx = blockIdx.x * blockDim.x + threadIdx.x;
> >     if (idx >= N) return;
> >     float a_val = a[idx];
> >     float b_val = b[idx];
> >     out[idx] = (a_val > b_val) ? a_val * a_val : b_val * b_val;
> > }
> > ```
> >
> > **Observation**: The baseline LLM replicates the branching logic from the original PyTorch code, thereby preserving the warp divergence issue.
> >
> > ### EG-CFG Generated CUDA
> >
> > ```cpp
> > __global__ void solution(const float* a, const float* b, float* out, int N) {
> >     int idx = blockIdx.x * blockDim.x + threadIdx.x;
> >     if (idx >= N) return;
> >
> >     float ai = a[idx];
> >     float bi = b[idx];
> >     float max_val = fmaxf(ai, bi);
> >     out[idx] = max_val * max_val;
> > }
> > ```
> >
> > **Optimization**: EG-CFG **removes the branch** by using `fmaxf`, a warp-friendly operation that selects the maximum value without divergence. This leads to improved warp coherence and better GPU utilization.
> >
> > ### Gemini 2.5 Pro Generated CUDA
> >
> > ```cpp
> > __global__ void solution(const float* a, const float* b, float* out, int N) {
> >     int idx = blockIdx.x * blockDim.x + threadIdx.x;
> >     if (idx >= N) return;
> >     float val = fmaxf(a[idx], b[idx]);
> >     out[idx] = val * val;
> > }
> > ```
> >
> > **Observation**: Gemini 2.5 Pro also generates a warp-friendly implementation that uses `fmaxf`. Structurally, its solution is functionally similar to EG-CFG’s output and avoids branching. However, it does not outperform EG-CFG in execution metrics; the quantitative analysis below reveals why.
> >
> >
> >
> > ### Quantitative Evidence
> >
> >
> > EG-CFG achieves the best overall performance score at −1.35, outperforming both the baseline LLM (−0.76) and Gemini 2.5 Pro (−1.06). All three models achieve the same executed IPC (−0.40 for EG-CFG and Gemini, −0.32 for the baseline), and both EG-CFG and Gemini reach an improved warp cycles per instruction of 0.50 compared to the baseline’s 0.33. This confirms that removing the branch helped reduce warp-level inefficiency.
> > The key difference lies in memory access efficiency. EG-CFG reaches a significantly lower L2 cache miss penalty (L2 hit rate z-score: −1.67), compared to −0.22 for Gemini and −0.15 for the baseline. Occupancy and SM busy scores are similar across all versions, but EG-CFG’s deeper optimization of memory patterns sets it apart.
> > ### Outcome
> >
> > EG-CFG delivers the largest performance gain in the example, improving over the baseline by 77%. It not only removes warp divergence like Gemini but also goes further by optimizing memory-level behavior. This example demonstrates how execution guidance enables the model to go beyond structural transformations and discover deeper architectural improvements that lead to real performance gains.

---

> > > ### Author Response · Authors · 2025-08-08
> > > **Non-Python Evaluation: CUDA Code Optimization via EG-CFG (part 4 -- second qualitative example)**
> > >
> > > ## Example 8
> > > we present another case, **example\_8** - where EG-CFG discovers a subtle yet impactful optimization opportunity in arithmetic simplification.
> > >
> > > ### PyTorch Function (Suboptimal)
> > >
> > > ```python
> > > def solution(a: torch.Tensor, b: torch.Tensor) -> torch.Tensor:
> > >     out = torch.zeros_like(a)
> > >     for i in range(a.shape[0]):
> > >         temp = a[i] * 0.1 + b[i] * 0.1
> > >         out[i] = temp + temp
> > >     return out
> > > ```
> > >
> > > **Issue**: This function performs **redundant scalar multiplications by 0.1**, and **misses opportunities for operation fusion**. Both `a[i]` and `b[i]` are multiplied by the same scalar before being added, and the result is then duplicated. A more efficient variant would restructure the arithmetic to reduce the total number of operations.
> > >
> > >
> > > ### Baseline LLM Generated CUDA
> > >
> > > ```cpp
> > > __global__ void solution(const float* a, const float* b, float* out, int N) {
> > >     int idx = blockIdx.x * blockDim.x + threadIdx.x;
> > >     if (idx >= N) return;
> > >     float temp = a[idx] * 0.1f + b[idx] * 0.1f;
> > >     out[idx] = temp + temp;
> > > }
> > > ```
> > >
> > > **Observation**: The baseline LLM preserves the original arithmetic structure, performing **two multiplications** per element followed by a duplication of the result. This structure offers **no improvement** over the PyTorch baseline.
> > >
> > >
> > > ### EG-CFG Generated CUDA
> > >
> > > ```cpp
> > > __global__ void solution(const float* a, const float* b, float* out, int N) {
> > >     int idx = blockIdx.x * blockDim.x + threadIdx.x;
> > >     if (idx >= N) return;
> > >
> > >     float temp = (a[idx] + b[idx]) * 0.1f;
> > >     out[idx] = temp + temp;
> > > }
> > > ```
> > >
> > > **Optimization**: EG-CFG restructures the expression by factoring out the common scalar multiplier and combining the operands first. This reduces the number of instructions and helps avoid unnecessary floating-point operations, ultimately resulting in a leaner kernel.
> > >
> > > ### Gemini 2.5 Pro Generated CUDA
> > >
> > > ```cpp
> > > __global__ void solution(const float* a, const float* b, float* out, int N) {
> > >     int idx = blockIdx.x * blockDim.x + threadIdx.x;
> > >     if (idx >= N) return;
> > >     out[idx] = 0.2f * (a[idx] + b[idx]);
> > > }
> > > ```
> > >
> > > **Observation**: Gemini Pro 2.5 reaches the same algebraic transformation as EG-CFG, but its implementation lacks the same hardware-level efficiency improvements, particularly in memory behavior.
> > >
> > >
> > > ### Quantitative Evidence
> > >
> > > EG-CFG achieves the best overall performance score at −1.28. This improves on the baseline LLM’s score of −1.17 and Gemini’s −1.06. Executed IPC is identical for both EG-CFG and the baseline at −0.32, while Gemini operates at a slightly lower −0.40. EG-CFG also improves warp cycles per instruction, increasing from 0.30 in the baseline to 0.55, indicating more efficient arithmetic throughput. Gemini matches EG-CFG with a warp cycles per instruction score of 0.50.
> > >
> > > The key difference again lies in memory behavior. EG-CFG significantly improves the L2 hit rate, reaching −1.58, compared to −2.22 for the baseline and −0.12 for Gemini. Occupancy and SM busy metrics remain similar across all three variants. These improvements suggest that the arithmetic simplification in EG-CFG not only reduces instructions but also contributes to better memory access patterns.
> > >
> > >
> > > ### Outcome
> > >
> > >
> > > EG-CFG improves over the baseline by 9.3 percent in performance score for this task. Although the gain is smaller than in example 1, this case shows the method’s ability to catch and simplify small inefficiencies in arithmetic logic. Even when the original code looks simple, EG-CFG restructures it to avoid extra operations and make execution more efficient. This highlights how the method can improve both major issues and small details in the code
> > >
> > > To summarize, while these are still preliminary experiments, this compelling use case underscores both the robustness and generality of our method.

---

### Note · Authors · 2025-08-12

We thank the AC and the reviewers for their effort and helpful feedback. Based on the comprehensive discussion with reviewers, we have addressed all concerns raised and demonstrated the strength of our work through extensive additional experiments.

Initially evaluated only on MBPP, we responded to reviewers' requests for broader evaluation by expanding to six major benchmarks, achieving state-of-the-art results across all: MBPP (96.6%), MBPP-ET (73.0%), HumanEval (99.4%), HumanEval-ET (89.0%), CodeContests (60.6%), and DS-1000 (69.9%).

To address computational fairness concerns (7xf3, GpF1, FLp2), we demonstrated that our method's advantage isn't merely from higher token usage. We ran baseline methods with 40x more retries/tokens, achieving only marginal improvements (6.8% for MGDebugger, 1.6% for MapCoder), still far below our performance. Moreover, 5/6 benchmarks contain hidden test cases, preventing overfitting to public tests.

Responding to reviewer fegN's request for non-Python evaluation, we developed a novel CUDA optimization benchmark during rebuttal. EG-CFG generates CUDA kernels that are 36% more efficient than baseline LLM and 31% better than Gemini 2.5 Pro.

We clarified methodological questions: the outer loop behavior, execution trace format, beam search implementation, and termination conditions. We added pseudocode and examples as requested. Reviewer GpF1's concern about context window limitations was addressed by showing our method handles 1k-line functions within modern LLM context windows.

Those reviewers who engaged with our responses reacted positively. GpF1 raised their score after clarifications, stating "main concerns have been addressed." FegN acknowledged our "comprehensive rebuttal" and "consistent performance gains across a large number of evaluations," maintaining an acceptance score, even before seeing the CUDA results that address their remaining concern.. While reviewers FLp2 and 7xf3 haven't responded to our rebuttal yet, we've thoroughly addressed their concerns through extensive new experiments and clarifications.

The unanimous recognition of our method's novelty, combined with our comprehensive empirical validation, demonstrates both the technical contribution and practical impact of EG-CFG. We've shown it's not limited to simple Python tasks but generalizes to complex competitive programming, data science, and even GPU optimization tasks, establishing it as a significant advancement in neural code generation.

---

### Decision · Program_Chairs · 2025-09-17

**Decision:**

Accept (poster)

**Comment:**

This paper proposes Execution-Guided Classifier-Free Guidance (EG-CFG), an inference-time decoding method for code generation that integrates real-time execution feedback into the generation process. The approach is conceptually novel, lightweight, and does not require additional training. It is inspired by how human programmers iteratively run partial code to guide development and achieves state-of-the-art performance on MBPP.

The method is original and clearly presented, combining execution feedback with classifier-free guidance in a simple, reproducible manner. Reviewers appreciated the paper’s clarity and the ablations, which substantiate the contributions of different components. Empirically, the approach demonstrates significant performance improvements, outperforming strong baselines.

The main concern is the limited evaluation scope. Experiments are restricted to MBPP, which lacks hidden test cases and is somewhat dated. This raises questions about generalizability to other benchmarks (e.g., LiveCodeBench, HumanEval, BigCodeBench) and to real-world programming tasks beyond Python. Some reviewers also noted that computational overhead and certain procedural details could be discussed more explicitly.

Despite these limitations, the paper presents a creative and impactful contribution. I lean toward acceptance, as the work is timely, methodologically sound, and likely to stimulate further research in inference-time program synthesis.